# You Cannot Feed Two Birds with One Score: the Accuracy-Naturalness Tradeoff in Translation

**Gergely Flamich**[*]  **David Vilar**  **Jan-Thorsten Peter**  **Markus Freitag**
Imperial College London  Google
g.flamich@imperial.ac.uk  {vilar, jtp, freitag}@google.com

## Abstract

The goal of translation, be it by human or by machine, is, given some text in a source language, to produce text in a target language that simultaneously 1) preserves the meaning of the source text and 2) achieves natural expression in the target language. However, researchers in the machine translation community usually assess translations using a single score intended to capture semantic accuracy and the naturalness of the output simultaneously. In this paper, we build on recent advances in information theory to mathematically prove and empirically demonstrate that such single-score summaries *do not and cannot* give the complete picture of a system's true performance. Concretely, we prove that a tradeoff exists between accuracy and naturalness and demonstrate it by evaluating the submissions to the WMT24 shared task. Our findings help explain well-known empirical phenomena, such as the observation that optimizing translation systems for a specific accuracy metric (like BLEU) initially improves the system's naturalness, while "overfitting" the system to the metric can significantly degrade its naturalness. Thus, we advocate for a change in how translations are evaluated: rather than comparing systems using a single number, they should be compared on an *accuracy-naturalness plane*.

## 1 Introduction

Machine translation (MT) is one of the undeniable success stories of computer science and machine learning research. Its history is long and varied, dating back to at least the 1960s and having progressed through symbolic, statistical and, most recently, neural approaches. By now, the performance of state-of-the-art systems is close to or matches that of human translators. But how do we assess the quality of translation systems? Indeed, how *should* we assess them? Already in the early days of MT, researchers recognized the need to evaluate translations along two axes: the ALPAC project considered the "intelligibility" and "fidelity" of the translations (Pierce & Carroll, 1966), while DARPA evaluations used "adequacy" and "fluency" (White & O'Connell, 1993). Although these concepts are intuitively appealing, they partly fell out of favour for two main reasons. First, they lack solid, widely-accepted formal definitions, which made it 1) difficult to define guidelines for human evaluations, and 2) impossible to establish their theoretical properties and their interaction. Second, in recent years, optimising solely some notion of accuracy where we compare an MT hypothesis with a given reference, has already lead to fluent translations in practice. Hence, even the relevance of such a distinction has become unclear. Nonetheless, using a single metric for automatic translation evaluation has always had issues. For example, for a few years now, it has been clear that classical "overlap metrics," such as BLEU and chrF are no longer good enough to assess translation quality, as they do not correlate well with human ratings, which prompted researchers to move towards neural metrics instead (Freitag et al., 2022a).

---

[*]Work done while a Student Researcher at Google as a doctoral student at the University of Cambridge. Now at Imperial College London.

**Note:** the paper title is a wordplay on "You cannot feed two birds with one scone," a more graceful and less lethal variant of a popular English saying.

However, in the latest WMT general task, a similar phenomenon occurred: systems with the best automatic scores (based on neural metrics) did not achieve the best score among human raters (Kocmi et al., 2024). This and related phenomena motivated us to reexamine translation evaluation practices.

**The accuracy-naturalness tradeoff.** As we note above, at present there is an implicitly held belief (or wishful thought) in the community that there should exist some metric, which can be optimized on to yield a translation system that is simultaneously accurate and natural-sounding. To examine the plausibility of this belief, we propose formal definitions of accuracy and naturalness by extending the mature, and widely-accepted information-theoretic framework of Blau & Michaeli (2018). Then, we show that under these definitions, the above goal is hopeless: there is **no metric**, optimizing which will yield a system that is simultaneously accurate and perfectly natural-sounding. Specifically, our contributions are:

- We extend Blau & Michaeli (2018)'s distortion-perception theory to translation, and mathematically prove and empirically demonstrate that there is a tradeoff between the accuracy of translations and their naturalness.
- We introduce target-only naturalness assessment, and establish a theoretical link between averaging these scores and approximating statistical distances.
- We approximate accuracy-naturalness curves using LLM-generated samples.
- Using our theory, we explain the above-mentioned phenomena when optimizing for accuracy metrics: while accuracy and naturalness correlate far from their Pareto-frontier, they begin to anti-correlate close to the frontier.

## 2 Background and Notation

**Assessing translation accuracy.** The primary measure of a translator's ability/quality is their *accuracy*, that is, how well their translations capture the meaning of the source text. Therefore, quantifying accuracy necessarily involves a comparison between the translator's output and a *reference*. In practice, the reference could be the source text, some translation produced by a professional translator that is taken to be ground truth, or both. In this paper, we refer to any metric that performs such a comparison as a *reference metric*. Classic examples include BLEU (Papineni et al., 2002) and chrF (Popović, 2015), which compare candidates with reference translations. They are convenient and simple, because they assess translations on a *lexical level* with the hope that proximity to the reference translation will ensure that the translations are correct at the *semantic level* as well. In recent years, the community has moved toward neural reference metrics, such as MetricX (Juraska et al., 2024) and COMET (Rei et al., 2020), which aim to assess translations directly at the *semantic level* instead. These metrics typically aim to directly predict human opinion scores from large language model (LLM) features. They still rely on reference translations, but are more robust and hence better suited for evaluating modern MT systems (Freitag et al., 2022a). As a further development, quality estimation (QE) metrics drop the need for a reference translation, but as they take the source text as input, we consider them reference metrics too.

In our paper, we identify *accuracy* with *reference metrics* and use the terms reference metric and accuracy metric interchangeably. Arguably, neural metrics measure a combination of adequacy and fluency, but in our framework they are nonetheless accuracy metrics because they still rely on a reference translation (or the source sentence in case of QE metrics).

**Assessing translation naturalness.** Producing fluent translations has always been an objective of translation systems. In the case of MT, early systems struggled to even produce grammatically correct sentences. As the quality of automatic systems improved, this became less of an issue, but the community encountered another problem, long known in the (human) translation community: translated text often does not sound completely natural, it contains *translationese* (Gellerstam, 1986). To study and eliminate this phenomenon, researchers have proposed several candidate metrics for naturalness, such as the language model marginal (log-)probability (Freitag et al., 2022b; Lim et al., 2024) as well as linguistically motivated metrics as proposed by Vanmassenhove et al. (2021). These scores are also called *no-reference metrics* as they do not rely on additional data. Note that no-reference

metrics are not to be confused with QE metrics, which we consider to be reference metrics as we explained above. However, although they are intuitive, no-reference metrics lack a more formal mathematical justification. In this paper, we develop an abstract notion of naturalness based on information theory, and show how no-reference metrics conform to our definitions, which provides a unifying framework.

**Terminology and notation.** We denote some randomly selected source text by the random variable $\mathbf{x} \in \mathcal{X}$ and some random target text by $\mathbf{y} \in \mathcal{Y}$, with $\mathcal{X}$ ($\mathcal{Y}$) the set of all possible source (target) texts. We denote probability distributions by the capital roman letters $P$, $Q$ and $R$, and denote the expectation of a function $f$ with respect to distribution $P$ as $\mathbb{E}_{\mathbf{x} \sim P}[f(\mathbf{x})]$. Source-target pairs are drawn from the joint distribution $P_{\mathbf{x},\mathbf{y}}$. This joint distribution also immediately defines the marginal over the source sentences $P_{\mathbf{x}}$ and marginal over the target sentences $P_{\mathbf{y}}$. For the purposes of this paper, we think of $P_{\mathbf{x}}$ as a "true" monolingual distribution over the source language and $P_{\mathbf{y}|\mathbf{x}}$ as the distribution of human translations into the target language given the source sentence $\mathbf{x}$. This philosophy reflects current practices in machine translation: the source language is typically derived from a monolingual corpus, which professional translators then translate into the target language. Indeed, there has been a recent explicit effort in the MT community to avoid back-translations or translations in both directions (Toral et al., 2018; Läubli et al., 2020).

Since we obtain $P_{\mathbf{y}}$ by marginalising over the input, it is the distribution over "human translations into the target language." Thus, it will be useful at times to consider a different distribution $R_{\mathbf{y}}$, which we can think of as a "true" monolingual reference over the target language. Finally, we model a translation system as a conditional distribution $Q_{\mathbf{y}|\mathbf{x}}$. Then, we can also immediately consider the "translator's marginal" $Q_{\mathbf{y}}(\cdot) = \mathbb{E}_{\mathbf{x} \sim P_{\mathbf{x}}}[Q_{\mathbf{y}|\mathbf{x}}(\cdot)]$.

## 3   The Accuracy-Naturalness Tradeoff in Translation

One of the key contributions of this paper is extending and generalising the work of Blau & Michaeli (2018), and applying it to translation. This section sets the scenery and then extends their theoretical results, so that they cover the scenarios we encounter in machine translation today. The key theoretical difference between their work and ours is that they develop their theory for data reconstruction. Our setting is fundamentally different: translation is not a reconstruction task, the input and output spaces are different.

**Reference metrics (accuracy).** We start by considering a direct transfer of the *distortion metrics* defined in Blau & Michaeli (2018). These are functions $\Delta : \mathcal{Y} \times \mathcal{Y} \to \mathbb{R}$ that take as input a reference translation $y^r$ and a candidate translation $y^c$ and output a real number that measures how close the candidate translation is to the reference translation. We require that: 1) $\Delta$ be bounded from below and 2) $\Delta$ is minimised when $y^c$ equals $y^r$. Lexical metrics like (negative) BLEU and (negative) ChrF clearly fulfill these conditions, but we need to expand this definition to accommodate modern MT evaluation metrics. First we note that some neural metrics (including QE metrics) take the source sentence as an additional "reference". We thus expand our functional form to $\Delta : \mathcal{X} \times \mathcal{Y} \times \mathcal{Y} \to \mathbb{R}$. The interpretation of the output value should also be generalized to the degree in which the candidate $y^c$ captures the *meaning* with respect to $x$ and $y^r$, and thus it is minimized when $y^c$ is semantically equivalent to the given reference. As such, a reference metric corresponds to a notion of *inaccuracy*. Current neural metrics are known or at least assumed to violate this requirement. For example, they might be vulnerable to "universal translations," which yield low distortion regardless of the reference (Yan et al., 2023). However, as we will see, our theory only requires the requirement to hold "on average," which is indeed the case for neural metrics. In most cases, the reference metrics the community uses are "accuracy-like" (higher is better), which can be considered just as a negated distortion, and thus we define the accuracy (or reference-score) of a translation system $Q_{\mathbf{y}|\mathbf{x}}$ as:

$$A(Q_{\mathbf{y}|\mathbf{x}}) = -\mathbb{E}_{\mathbf{x},\mathbf{y}^r \sim P_{\mathbf{x},\mathbf{y}}}[\mathbb{E}_{\mathbf{y}^c \sim Q_{\mathbf{y}|\mathbf{x}}}[\Delta(\mathbf{x},\mathbf{y}^r,\mathbf{y}^c)]] \qquad (1)$$

Virtually all commonly used translation metrics are designed to be reference metrics, they differ mainly on what arguments they utilize. For example, BLEU and ChrF ignore the

source, QE metrics ignore the target reference, and MetricX [1] or COMET use all three. We note that in the community there is an unwritten feeling that neural metrics place a heavier weight on how well a particular translated sentence sounds than on the accuracy of the translation. However, so long as there is an element of comparison with a reference, and assuming that our assumptions are satisfied, neural metrics are still reference metrics.

**No-reference metrics (naturalness).** Following Blau & Michaeli (2018), we argue that the naturalness of a translated text should be intrinsic to it, and should not depend on the source text. The basis of our thesis is the following desideratum: a set of translations ought to be **perfectly natural** if no observer/critic can reliably distinguish them from a monolingual corpus in the target language. More precisely, let $Q_{\mathbf{y}|\mathbf{x}}$ denote a (possibly and usually stochastic) translator. Then, we consider the "translator's marginal" $Q_{\mathbf{y}}(\cdot) = \mathbb{E}_{\mathbf{x} \sim P_{\mathbf{x}}}[Q_{\mathbf{y}|\mathbf{x}}(\cdot)]$, which is the output distribution we obtain when we randomly select an input $\mathbf{x} \sim P_{\mathbf{x}}$ and translate it: $\mathbf{y} \sim Q_{\mathbf{y}|\mathbf{x}}$. Then, we say that *a translation system $Q_{\mathbf{y}|\mathbf{x}}$ is perfectly natural with respect to a monolingual reference distribution $R_{\mathbf{y}}$ in the target language, if the system's marginal matches the reference:* $Q_{\mathbf{y}} = R_{\mathbf{y}}$. Crucially, it is up to the practitioner to define $R_{\mathbf{y}}$. Of course, one could set $R_{\mathbf{y}} = P_{\mathbf{y}}$ to the marginal defined by our parallel dataset; however, this is not necessary and indeed there are many good reasons not to! For example, the reference translations might contain "human translationese" (Gellerstam, 1986) and it is usually better to assess the quality of the system against text that was conceived in the target language. Furthermore, translation systems for different domains all require different notions of "naturalness" due, e.g., to the specific terminology that they use. We might also wish to select $R_{\mathbf{y}}$ such that its support is filtered for unwanted content, such as slurs/curse words.

In practice, requiring perfectly natural translations is too stringent and hence we relax it such that $Q_{\mathbf{y}} \approx R_{\mathbf{y}}$ instead, which we formalise using statistical distances. Concretely, we pick some *divergence $D(Q, R)$*, that is, $D(Q, R) \geq 0$ for any two distributions $Q, R$ and $D(Q, R) = 0$ if and only if $Q = R$. However, similarly to distortion, a statistical distance measures the *unnaturalness* or *disfluency* of a system $Q_{\mathbf{y}|\mathbf{x}}$. Hence, we define the naturalness with respect to a statistical distance $D$ and reference distribution $R_{\mathbf{y}}$ as the negated distance

$$N(Q_{\mathbf{y}|\mathbf{x}}) = -D(Q_{\mathbf{y}}, R_{\mathbf{y}}). \tag{2}$$

What distance $D$ should we choose? Natural choices of $D$ include integral probability metrics (IPM) such as the total variation and Wasserstein distances, and $f$-divergences, such as the Kullback-Leibler or Rényi divergences, as these families all have natural connections to distinguishability (Sriperumbudur et al., 2009); see Appendix A for a discussion.

Finally, we emphasize that a key feature of our theory is to recognize naturalness as a "corpus-/dataset-level" property. As such, the theory has no notion of "sentence-level" naturalness. In other words, our theory concerns itself with the naturalness of a translation system $Q_{\mathbf{y}|\mathbf{x}}$, but not with the naturalness of individual text segments.

### 3.1 The Tradeoff

Having defined accuracy and naturalness, we now study their interaction. First, observe that the two objectives are distinct: we can construct a system that is perfectly natural, but highly inaccurate by setting $Q_{\mathbf{y}|\mathbf{x}} \leftarrow R_{\mathbf{y}}$. Such a "translation system" ignores its input and outputs some randomly chosen text from the reference distribution $R_{\mathbf{y}}$. It would be a system with perfectly natural outputs, but utterly useless.

However, if we have a system that is optimally accurate, can we ever expect it to be perfectly natural? To answer this, consider neural metrics, such as MetricX or COMET, which are *trained* so that they implicitly jointly assess adequacy and fluency (in the MT sense). Concretely, let $\Delta^*$ be a neural metric we obtain by optimizing some appropriate objective. Now, consider a translation system $Q_{\mathbf{y}|\mathbf{x}}^*$ that minimizes $\Delta^*$, i.e., $\Delta(Q_{\mathbf{y}|\mathbf{x}}^*) = \min_{Q_{\mathbf{y}|\mathbf{x}}} \Delta^*(Q_{\mathbf{y}|\mathbf{x}})$. By definition, $Q_{\mathbf{y}|\mathbf{x}}^*$ is optimally accurate, but when is it also perfectly natural? Letting $Q_{\mathbf{y}}^*(\cdot) = \mathbb{E}_{\mathbf{x} \sim P_{\mathbf{x}}}[Q_{\mathbf{y}|\mathbf{x}}^*(\cdot)]$, we see that by our definition, $Q_{\mathbf{y}|\mathbf{x}}^*(\cdot)$ is perfectly natural when

---

[1] In their latest version.

$D(Q^*_\mathbf{y}, R_\mathbf{y}) = 0$, which occurs if and only if $Q^*_\mathbf{y} = R_\mathbf{y}$! Due to the flexibility of our framework, this allows us to *construct* a perfectly natural system from a perfectly accurate one: we just need to set the monolingual reference distribution $R_\mathbf{y} \leftarrow Q^*_\mathbf{y}$! However, this choice makes $R_\mathbf{y}$, and hence our notion of naturalness depend on $P_{\mathbf{x,y}}$, which we argue is problematic for assessing translation naturalness. Our philosophy is that the notion of naturalness in a given language $\mathcal{Y}$ should not depend implicitly or explicitly on another language $\mathcal{X}$. Otherwise, we arrive at the absurd conclusion that the naturalness in language $\mathcal{Y}$ is different when translating from language $\mathcal{X}_1$ into $\mathcal{Y}$ and when translating from language $\mathcal{X}_2$ into $\mathcal{Y}$! However, $Q^*_\mathbf{y}$ breaks our principle in two ways: 1) it depends on $Q^*_{\mathbf{y|x}}$, which in turn depends on $P_{\mathbf{x,y}}$ and 2) it depends the marginalisation. On the other hand, setting the reference distribution $R_\mathbf{y}$ to anything other than $Q^*_\mathbf{y}$ derived from a minimizer of $\Delta^*$ means that $Q^*_{\mathbf{y|x}}$ will not be perfectly natural. This argument illuminates that if we follow the philosophy that the monolingual reference $R_\mathbf{y}$ should not depend on $P_{\mathbf{x,y}}$, then the degenerate case of perfect accuracy implying perfect naturalness is exceedingly unlikely in practice. However, we emphasize that this conclusion relies on explictly disallowing $R_\mathbf{y}$ to depend on $P_{\mathbf{x,y}}$, the motivation for which comes from the theory's application to translation. In some other scenarios allowing such dependence might be sensible: for example, in Appendix B, we generalise Theorem 1 of Blau & Michaeli (2018), which does allow $P_\mathbf{y}$-dependence and admits a similar "no-two-birds-with-one-scone" result.

The above argument guarantees that there is almost always a tradeoff between accuracy and naturalness, but gives no estimate of its severity. Indeed, we might expect the tradeoff to be mild for language pairs with an almost one-to-one correspondence, such as Spanish-Catalan or Serbian-Croatian. On the other hand, the tradeoff should be more pronounced for more distant language pairs such as German-Japanese or Hungarian-Swahili. We quantify these tradeoffs in Section 5, specifically in Figure 3. Hence, to study the tradeoff, we define the *accuracy-naturalness* (AN) function[2]

$$A(N) = \max_{Q_{\mathbf{y|x}}} \{-\Delta(Q_{\mathbf{y|x}})\} \quad \text{subject to } -D(Q_\mathbf{y}, R_\mathbf{y}) \geq N. \tag{3}$$

$A(N)$ represents the best accuracy we can ever expect for a translation system, subject to achieving a naturalness of at least $N$. We have the following result, analogous to Theorem 2 of Blau & Michaeli (2018); see Appendix C for the proof.

**Theorem 3.1.** *Let $A(N)$ be the accuracy-naturalness function defined by distortion $\Delta$, distributional distance $D$, parallel distribution $P_{\mathbf{x,y}}$ and monolingual reference $R_\mathbf{y}$. Then:*

1. *$A(N)$ is **non-increasing** in N.*

2. *if D is convex in its first slot, then $A(N)$ is **concave** in N.*

That $A(N)$ is non-increasing in $N$, taken together with our argument above that a tradeoff exists for virtually all practically relevant cases implies that close to the curve, accuracy and naturalness **must anti-correlate**. That is, an increase in the naturalness of a system beyond a certain point must come at the sacrifice of some accuracy, and vice versa. The concavity of $A(N)$ in $N$, on the other hand, means that we can expect a larger and larger degradation in naturalness for a small gain in accuracy and vice versa, especially at the extremes.

**Example.** To make the inevitability of the tradeoff more concrete, consider translating the German sentence "Meine Lehrerin ist nett" to English. A sensible translation would be "My teacher is nice," which sounds natural, but loses the information from the German sentence that the teacher is female. An alternative translation could be "My female teacher is nice," which is now fully accurate, but sounds quite unnatural in English. Another, naturally occurring example of this phenomenon is the translation of Japanese honorifics. A *completely accurate* translation into English would need to introduce diverse paraphrases denoting the social relation between the speakers, which would result in unnatural text. This type of strict semantic differentiation is currently not explicitly assessed by any metric that we are aware of, but might be relevant in certain scenarios.

---

[2]The accuracy-naturalness function is related to the better-known distortion-perception function $\delta(P) = \min_{Q_{\mathbf{y|x}}} \{\Delta(Q_{\mathbf{y|x}})\}$ s.t. $D(Q_\mathbf{y}, R_\mathbf{y}) \leq P$ (Blau & Michaeli, 2018) via $\delta(P) = -A(-P)$.

## 4 Translation evaluation in practice

Before moving to the empirical analysis, we lay out the basis for a practical evaluation pipeline. Thus, let $\mathbf{X} = \{\mathbf{x}_1, \mathbf{x}_2, \ldots, \mathbf{x}_N\}$ and $\mathbf{Y}^{ref} = \{\mathbf{y}_1, \mathbf{y}_2, \ldots, \mathbf{y}_N\}$ be a parallel test set, with source-reference pairs drawn iid: $\mathbf{x}_i, \mathbf{y}_i \sim P_{\mathbf{x},\mathbf{y}}$. Furthermore, let $R_\mathbf{y}$ be a monolingual reference distribution in the target language, which we could obtain by estimating it from some monolingual corpus. Finally, we consider a set of $M$ translation systems $\mathcal{Q} = \{Q^1_{\mathbf{y}|\mathbf{x}'} \ldots Q^M_{\mathbf{y}|\mathbf{x}}\}$ to be evaluated. Following standard practice, we wish to compare models based on their outputs. To this end, we require that each system $Q^m_{\mathbf{y}|\mathbf{x}} \in \mathcal{Q}$ translates the source texts in the test set and produces a translation set $\mathbf{Y}^m = \{\mathbf{y}_n \sim Q^m_{\mathbf{y}|\mathbf{x}_n} \mid \mathbf{x}_n \in \mathbf{X}\}$. Our goal then is to estimate the accuracy and naturalness of each system using $\mathbf{X}$, $\mathbf{Y}^{ref}$, $R_\mathbf{y}$ and the $\mathbf{Y}^m$s.

### 4.1 No-reference metrics as statistical distances

So far, we followed a bottom-up approach, and developed our theory of accuracy and naturalness from the ground up. In this section, we switch to a top-down view, and ask whether current evaluation practices for assessing naturalness conform to our theory. The prevailing practical approach for data quality assessment are *no-reference metrics*. These can be either human-generated, such as the fluency scores derived from MQM assessments for text[3] (Freitag et al., 2021), or automated, such as the perplexity of a language model for text. We formalise no-reference metrics by assuming we have a set of $K$ critics $\mathcal{C} = \{f_1, f_2, \ldots f_K\}$ (usually with $K = 1$), each of which assigns a score to each set of translations $\mathbf{Y}^m$. Then, for each $f_k \in \mathcal{C}$, we compute the score $s^m_k = N^{-1} \sum_{\mathbf{y} \in \mathbf{Y}^m}^N f_k(\mathbf{y})$ and average the result to get the final score $s^m = K^{-1} \sum_{k=1}^K s^m_k$. Within the context of MQM assessment, the critics could be the set of human raters evaluating the output of a translation system. Then, $f_k(\mathbf{y})$ could be the $k$th rater's fluency score for the hypothesis $\mathbf{y}$ in $\mathbf{Y}^m$ derived from their MQM evaluation according to the conversion scheme of Freitag et al. (2021). Does this evaluation procedure fit our definition on naturalness? If so, how? As an answer to this question we propose the following statistical distance between two distributions $Q$ and $R$:

$$D_1(Q, R \mid \mathcal{P}) = \mathbb{E}_{f \sim \mathcal{P}} \left[ \left| \mathbb{E}_{Y \sim Q}[f(Y)] - \mathbb{E}_{Y \sim R}[f(Y)] \right| \right] \tag{4}$$

where $\mathcal{P}$ is a probability distribution over critics (technically, it is a stochastic process). Note the similarity of Equation (4) to integral probability metrics (Müller, 1997): instead of maximising, we average over critics instead; see Appendix A for a detailed discussion. Formally defining $D_1$ is somewhat technical, hence we relegate the details of the definition and the analysis of basic properties to Appendix E. Here we argue that averaging no-reference scores is, in fact, a rough Monte Carlo estimate of an appropriately chosen $D_1$. Indeed, assuming that the critic scores are all non-negative and are such that the reference distribution $R_\mathbf{y}$ always achieves a perfect score for any $f \sim \mathcal{P}$, that is, $\mathbb{E}_{Y \sim R_\mathbf{y}}[f(Y)] = 0$, then for a test set $\mathbf{Y}^m$ we have $D_1(Q^m_\mathbf{y}, R_\mathbf{y} \mid \mathcal{P}) \approx (KN)^{-1} \sum_{k=1}^K \sum_{\mathbf{y} \in \mathbf{Y}^m}^N f_k(\mathbf{y})$ with $f_k \overset{iid}{\sim} \mathcal{P}$, which is precisely the score $s$ we obtain from averaging no-reference metrics.

## 5 Experiments

In this section, we empirically verify our theory. We based all our experiments on publicly available data: the submissions and human MQM evaluations of the WMT24 general task[4].

### 5.1 Approximating the Accuracy-Naturalness curve using LLMs

First, we aim to get a basic feel for the accuracy-naturalness curve. Unfortunately, performing the optimisiation in Equation (3) is hopeless. Hence, we approximate it in the

---

[3]Strictly speaking, MQM evaluation of fluency is not completely reference-free, as the judges have access to the source sentence. But fluency-related scores are judged mainly on a monolingual level.

[4]Available at https://github.com/wmt-conference/wmt24-news-systems.

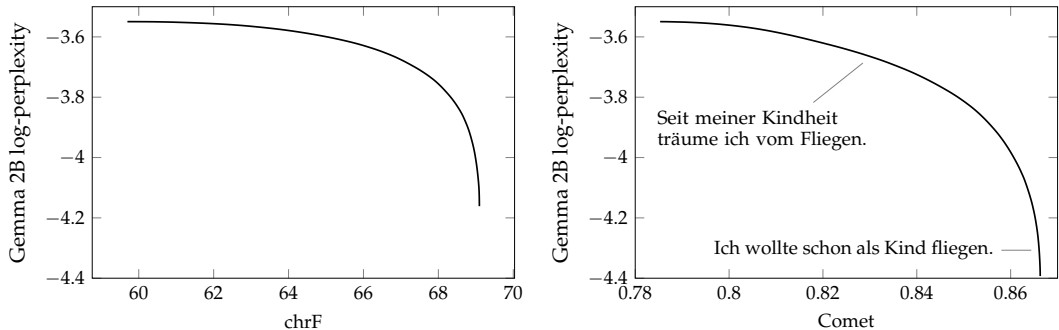

Figure 1: Approximation of the accuracy naturalness curve on the WMT24 en → de test set. **Left:** chrF vs LM log-perplexity. **Right:** COMET vs LM log-perplexity. To further illustrate the tradeoff, we add selected translations for the test sentence "I've wanted to fly since I was a child." The translation with higher COMET score is a fully accurate translation of the sentence, while the other translation is less accurate but more fluent according to our metric.

following way: using Gemini 1.5 Flash (Gemini Team et al., 2024), we generated $K = 1024$ candidate translations $\{\mathbf{y}_i^k\}_{k=1}^K$ for each source sentence $\mathbf{x}_i$ in the WMT24 en → de test set; see Appendix F for the precise details. Next, we chose to two accuracy metrics $\Delta$ to test for: chrF (Popović, 2015) and COMET (Rei et al., 2020). For the naturalness metric, we used the pretrained Gemma2 2B model log-perplexities (Gemma Team et al., 2024). Concretely, let $\Gamma(y)$ denote the probability of a sequence of tokens $y$ as predicted by the Gemma2 model, let $|y|$ denote the number of tokens in $y$ and hence let $\text{LPP}(Q) = \mathbb{E}_{Y \sim Q}\left[-\log \Gamma(Y)/|Y|\right]$ denote the log-perplexity of a distribution $Q$ with respect to $\Gamma$. Then, our concrete instantiation of $D_1$ is $D(Q, R) = |\text{LPP}(Q) - \text{LPP}(R)|$. As we show in Appendix E.1.2, this quantity can also be viewed as a more general version of the $D_1$ distance. We estimated the monolingual target reference distribution $R_{\mathbf{y}}$, using 20,000 entries of the 2024 NewsCrawl DEU Monolingual dataset[5]. Of course this choice for $R_{\mathbf{y}}$ represents only a specific type of language (that of news text), and does not represent "colloquial German" for example. Depending on the application, $R_{\mathbf{y}}$ could be adapted by using a different dataset; in our case it merely serves an illustrative purposes In our experiments, we found that $\text{LPP}(R_{\mathbf{y}})$ was always smaller than $\text{LPP}(Q_{\mathbf{y}}^m)$, for each system we examined, hence $D$ is equivalent up to an additive constant to the average model log-perplexity on the test dataset $D(Q_{\mathbf{y}}, R_{\mathbf{y}}) = \text{LPP}(Q_{\mathbf{y}}) + \mathcal{O}(1)$. To optimise Equation (3), we first construct the objective's "one-shot Lagrange dual" with tradeoff parameter $\beta$:

$$\mathcal{L}(x, y, y', \beta) = -\Delta(x, y, y') + \frac{\beta}{|y'|} \log \Gamma(y'). \tag{5}$$

Then, for each entry $\mathbf{x}_i$ in the WMT test set, we computed the optimal (aka oracle) translation for a large range of betas $\beta \in [10^{-4}, 10^4]$: $\mathbf{y}_i^* = \arg\max_{k \in [1:K]} \mathcal{L}(\mathbf{x}_i, \mathbf{y}_i, \mathbf{y}_i^k, \beta)$. Finally, we report the average of the metrics over the best translations selected:

$$\Delta^* = \frac{1}{N} \sum_{i=1}^N \Delta(\mathbf{x}_i, \mathbf{y}_i, \mathbf{y}_i^*), \qquad D^* = -\frac{1}{N} \sum_{i=1}^N \frac{1}{|\mathbf{y}_i^*|} \log \Gamma(\mathbf{y}_i^*). \tag{6}$$

We plot the results in Figure 1 and see that it verifies our theory: 1) there is indeed a tradeoff curve (see also Theorem B.1) and 2) the curve is non-increasing and concave, as Theorem 3.1 predicts. Finally, note that curves produced this way *overestimate* the true AN function and hence are overly optimistic, see Appendix F for the details.

### 5.2 Evaluating existing translation systems using the accuracy-naturalness framework

The plots of the previous section depict a simulation of the tradeoff curve through a controlled experiment, but how do real systems behave with respect to the ideal curve? In

---

[5]Available at https://data.statmt.org/news-crawl/de/.

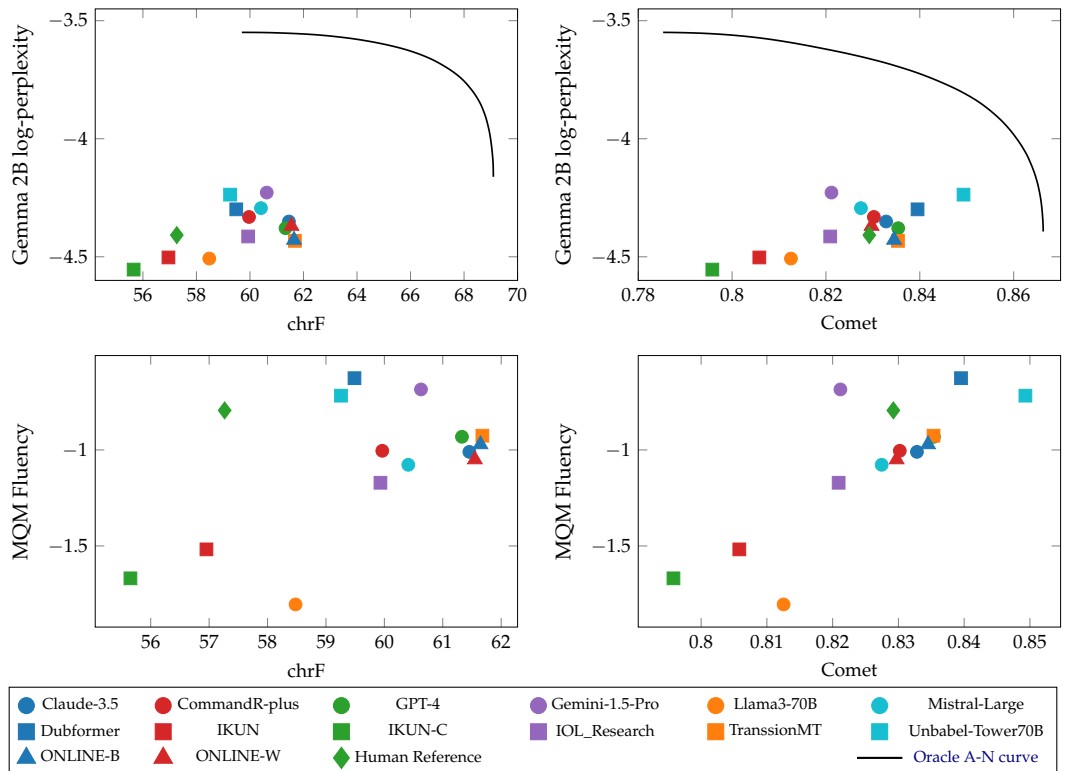

Figure 2: System performances on the WMT24 en → de test set. ● marks represent LLMs, ■ marks represent systems specifically trained for MT, ▲ marks represent "online" systems and the ♦ represents the performance of an alternative human reference. For further explanation of these categories, please see Kocmi et al. (2024). **Top row:** "Fully automatic" accuracy-naturalness comparisons using chrF and Comet as accuracy metrics and Gemma2B log-perplexity (see section 5.1) as the naturalness metric. We also include the approximate AN curves from Figure 1, though note that it is an optimistic estimate of the real curve. **Bottom row:** comparisons of automatic accuracy metrics (chrF and Comet) against human MQM fluency ratings, as described in section 5.2.

the top row of Figure 2, we include points for the systems that participated in the shared task in addition to the previously computed curve. All systems lie below the curve, as expected. Furthermore, the best performing systems in the evaluation (according to human judgement) are the ones that are closer to the curve. In order to investigate this assertion in more detail, in the bottom row of Figure 2 and in Figure 3 we compare human fluency scores against automated accuracy metrics and human adequacy scores[6] for those languages for which MQM evaluations are available. Note that in this case we cannot compute (or simulate) the tradeoff curve for human judgements.

We can see different aspects that confirm our theory. A first observation is that for "low-performing" systems (the ones farther away from the (0,0)-origin, which would represent optimal performance), accuracy/adequacy and fluency do correlate, but as the quality of the systems improve ("as we come closer to the curve") there is an anticorrelation. From comparing the plots in the bottom row of Figure 2, we see that Comet, a neural metric, correlates with fluency "for much longer," that is, its accuracy-naturalness tradeoffs should be much milder. Nonetheless, our theory shows that the tradeoff exists here too, and therefore at some point these will begin to anti-correlate with fluency as well. The underlying phenomenon is the same one that explains why BLEU (a pure accuracy metric)

---

[6]See Appendix D for details on the computation of fluency and adequacy scores from MQM evaluations.

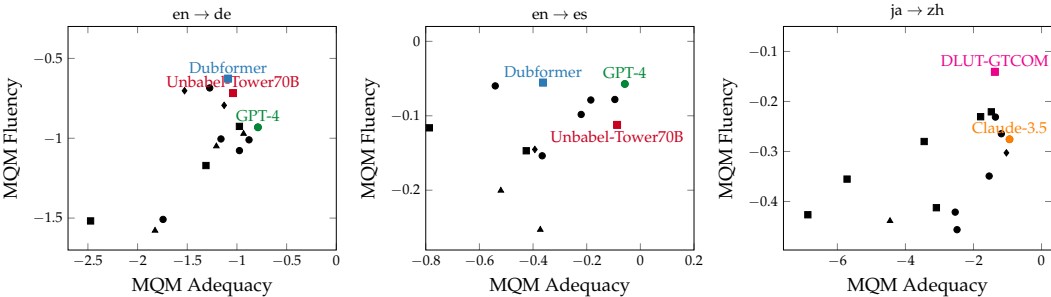

Figure 3: Plot of the accuracy-naturalness tradeoff derived from human MQM adequacy and fluency ratings, as described in Section 5.2. We highlight the top-performing systems at the Pareto frontier.

is no longer an adequate metric for system performance. We also observe a similar effect for different language pairs in Figure 3: the en→de and ja→zh systems exhibit a more pronounced accuracy-naturalness trade-off, while it is milder for en→es systems.

Secondly, we focus on interpreting the results for the performance of single systems. Unbabel (Rei et al., 2024) had the dominant system in the automatic metrics, but in human evaluations other systems were judged better. Unbabel's system was clearly optimized for COMET, an accuracy measure. Looking at the plots in Figure 3 we can see that the Unbabel system has a quite strong adequacy score, but due to the tradeoff the fluency of the system suffers. Other systems that human raters judged to be better lie closer to the tradeoff curve.

## 6 Related Work

Already in the early days of machine translation the need of evaluating across two dimensions was recognized (Pierce & Carroll, 1966; White & O'Connell, 1993). These were also the preferred metrics for many of the NIST MT evaluation in the early 2000s until there was a shift to single HTER scores (Snover et al., 2009; Habash et al., 2011). A similar trend can be found in the WMT evaluations. Whereas in the first editions adequacy and fluency was used (Koehn & Monz, 2006), this approach was quickly dropped in favor of ranking approaches (Callison-Burch et al., 2008) and later a single direct assessment score was given to each segment (Bojar et al., 2016). The use of MQM scores (Kocmi et al., 2022) allows for a distinction into adequacy and fluency evaluation, as we have done in this paper, but these type of evaluation is not available for all conditions and systems. All usual automatic metrics for machine translation evaluation are accuracy metrics as defined in Section 3. To the best of our knowledge there is not much work around intrinsic automatic metrics for the evaluation of the fluency of translations. Vanmassenhove et al. (2021) propose a series of statistical measures and show that they show different characteristics for machine translated and human produced texts, but these do not constitute statistical distances as we require here. There has been more attention into classifying translations into naturally occuring texts or translationese, which could be considered a measure of naturalness e.g. (Kurokawa et al., 2009; Lembersky et al., 2012; Freitag et al., 2022b).

Similar to our work, Lim et al. (2024) also recently identify a tradeoff between adequacy and fluency, in their case identifying adequacy with the conditional and fluency with the prior probabilities in a source-channel formulation of the translation problem. Our approach is more general, as we show that the tradeoff exists for every possible accuracy metric considered. They also argue for reintroducing explicit adequacy and fluency guided evaluations, as we do. Orthogonally, Elangovan et al. (2024) recently established a basis for claiming whether a particular automatic metric represents human judgements well.

As we have noted on several occasions, our work is heavily inspired by the work of Blau & Michaeli (2018). Before its publication, the neural image restoration and compression community was grappling with the analogous issue and widely held the same miscon-

ception the MT community holds now: despite developing ever-more sophisticated, so called "perceptual distortion metrics" that were inspired by/based on the human visual system, and which correlated well with human judgments, systems optimized using these metrics inevitably showed undesirable visual artefacts. The community's response to this was that "we just haven't found the right metric yet," and spent considerable effort on solving this issue. Blau and Michaeli's work showed that this effort is hopeless, and since then evaluating the performance of image reconstruction and compression algorithms along the distortion-perception plane has become standard practice. We hope our work will help bring about a similar change in the MT community.

The original idea has since been significantly extended and analysed in the context of data compression, see Blau & Michaeli (2019); Theis & Wagner (2021); Chen et al. (2022). We extend the framework by considering the translation setting and allowing to choose the reference distribution $R_{\mathbf{y}}$. One important aspect we left mostly unexplored, but which is an important next step once the tradeoff is recognised, is optimising it directly. In the image compression community, this has been recognised for a while, with many state-of-the-art compression models trained using a mixture of accuracy and naturalness (via adversarial losses), see e.g., HiFiC (Mentzer et al., 2020). However, adversarial approaches have recently been far less prominent in the text generation space, due to the difficulty in back-propagating through the generated text. Indeed, finding even the right framework for optimising no-reference metrics is an open problem (Theis, 2024). In the field of machine translation and text generation there has been some work on optimizing several objectives concurrently, e.g. (Duh et al., 2012; Kumar et al., 2021). Similar frameworks can potentially be used to optimize for a specific tradeoff of adequacy and fluency.

## 7 Discussion

In this paper, we showed that there is an inherent tradeoff in the accuracy and the naturalness of translations in virtually any practical scenario. We achieved this by extending and generalising Blau & Michaeli (2018)'s distortion-perception tradeoff to translation. Furthermore, we established a connection between no-reference metrics and statistical distances, which not only justifies our approach, but also the use of metrics such as those used on Blau & Michaeli (2018) and popular no-reference metrics used in image reconstruction such as LPIPS (Zhang et al., 2018). We empirically demonstrated the tradeoff by approximating the true AN curve using LLM-generated candidates and computing oracle scores on them. Finally, we showed how the accuracy naturalness tradeoff can bring more light into the interpretation of the results of evaluation campaigns, by plotting the submissions to the WMT24 general shared task on the accuracy-naturalness plane with respect to both automatic and human MQM-based accuracy and naturalness metrics.

Modern neural MT metrics pose an interesting question with respect to our theory. As they use either reference translation or the source sentence for comparison when judging candidate translations, they are instances of accuracy metric as we have defined them in this paper. But the clear consensus in the community is that they also have a strong fluency component. Our work can help to provide a better interpretation as to what these metrics are judging, and what effect optimizing for them has on the performance of MT systems.

Although this paper is mainly theoretical in nature, we posit that it can have important consequences in practice. Our first and foremost goal is to make the community aware of this tradeoff. We suggest to expand future evaluations to explicit consider again a distinction between accuracy and fluency (or similar dimensions) when evaluating MT systems.

Similar as in the evolution seen in the image reconstruction community, translation systems can be optimized and evaluated with respect to this tradeoff. Different applications may have different optimal points along the curve: while factual texts (e.g. scientific or legal documents) naturally favor adequacy, literary translations may benefit from a more natural flow of the produced text. Having the possibility of steering the behaviour of the systems can improve their flexibility and improve the final quality of translation systems.

## Reproducibility

The experimental section of this paper is mostly based on publicly available data produced in the WMT24 evaluation campaign. The biggest source of variability is sampling of Gemini, which by its own nature is a stochastic process, and has an additional dependence on the actual version of the model. However, as we are studying the general behaviour of the curve, rather than the actual values, the conclusions will still be valid for different samples or model updates.

## Acknowledgements

We thank Eleftheria Briakou for her detailed comments that helped improve an earlier version of our manuscript. We also thank Colin Cherry, Géza Kovács, Dan Deutsch, Arthur Gretton and Deniz Gündüz for fruitful discussions that helped us formulate our arguments more clearly and effectively.

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

# Appendices

## A  Selecting distributional divergences

There are many mathematically natural candidates available, such as total variation, the Wasserstein distances, Kullback-Leibler or other $f$-divergences and so on. A particularly appealing class of metrics to consider are the integral probability metrics (IPMs; Müller, 1997). For convenience, we introduce the shorthand $P(f) = \mathbb{E}_{X \sim P}[f(X)]$. Then, IPMs are defined as

$$\text{IPM}_{\mathcal{F}}[Q\|P] = \sup_{f \in \mathcal{F}} |Q(f) - P(f)|, \tag{7}$$

where the precise IPM is defined by the function class $\mathcal{F}$ we choose to optimise over. IPMs include the total variation distance, Wasserstein distance and maximum mean discrepancy (MMD; Gretton et al., 2012). An appealing aspect of IPMs is their direct connection to the ability of the optimal observer / critic (as a member of the function class $\mathcal{F}$) to distinguish $Q$-samples from $P$-samples (Sriperumbudur et al., 2009). It is useful to consider that in many cases, we can consider an IPM as the separation achieved by the *optimal critic*: if we consider $f^* = \arg\max_{f \in \mathcal{F}} |Q(f) - P(f)|$, then $\text{IPM}_{\mathcal{F}}[Q\|P] = |Q(f^*) - P(f^*)|$.

This viewpoint reveals the main practical issue with IPMs, which was already noted for $f$-divergences by Theis (2024): the optimal critic depends on the distributions $Q$ and $P$ that we are trying to evaluate. One solution that is often employed in lossy data compression is therefore to train the critic alongside the the generative model, leading to an adversarial training framework (Mentzer et al., 2020; Theis, 2024). The advantage of this approach is that besides obtaining a critic for evaluating the model at test time, it allows us to optimise our models using the accuracy-naturalness objective in Equation (5) directly. Unfortunately, due to the discrete nature of text outputs, adversarial approaches of text generation have lagged behind autoregressive approaches (Caccia et al., 2020), hence more research is needed before translation models can be trained using the analogous accuracy-naturalness objective. Hence, we focus on metrics that do not require optimisation: no-reference metrics. Following Section 4, we will be interested in quantities that can be estimated using the reference distribution $R_{\mathbf{y}}$ and the system outputs $\mathbf{Y}^m$.

### A.1  Using LLM model scores

Assume that we have "distilled" the true marginal into a language model $R_{\mathbf{y}} = P_{\mathbf{y}}^{LM}$. This simplifies the problem to assessing the distance between $Q_{\mathbf{y}}^i$ and $P_{\mathbf{y}}^{LM}$. This should be easier: let $f$ be the probability mass function of $P_{\mathbf{y}}^{LM}$. Then, we can also compute the probability $f(\mathbf{y})$ of the output text $\mathbf{y}$ under the language model. Then, we might be tempted to compute the perceptual scores as

$$C^m = \frac{1}{N} \sum_{j=1}^{N} -\log f(\mathbf{y}_j^m). \tag{8}$$

However, this is problematic, because, as we can see by the law of large numbers:

$$C^m \to \mathbb{E}_{\mathbf{y} \sim Q_{\mathbf{y}}^i}[-\log f(\mathbf{y})] \tag{9}$$

$$= \mathbb{H}[Q_{\mathbf{y}}^m \| P_{\mathbf{y}}^{LM}] \tag{10}$$

$$= D_{\text{KL}}[Q_{\mathbf{y}}^m \| P_{\mathbf{y}}^{LM}] + \mathbb{H}[Q_{\mathbf{y}}^m], \tag{11}$$

where $\mathbb{H}[Q\|P]$ is the cross-entropy of $Q$ from $P$. The issue here, is that $\mathbb{H}[Q_{\mathbf{y}}^m]$ could differ wildly among the translation systems, and hence the $C^m$ scores are incomparable! A different perspective on this problem is to note that the minimizer of the cross-entropy is in

general not $P_{\mathbf{y}}^{LM}$ (unless $P_{\mathbf{y}}^{LM}$ is deterministic, which is uninteresting):

$$Q_{\mathbf{y}}^* = \underset{Q_y}{\arg\min}\, \mathbb{H}[Q_y \| P_{\mathbf{y}}^{LM}] \tag{12}$$

$$= \underset{Q_{\mathbf{y}}}{\arg\min}\{D_{\mathrm{KL}}[Q_{\mathbf{y}} \| P_{\mathbf{y}}^{LM}] + \mathbb{H}[Q_{\mathbf{y}}]\} \tag{13}$$

Intuitively, $Q_{\mathbf{y}}^*$ balances proximity to $P_{\mathbf{y}}^{LM}$ (due to the KL term) and diversity (due to the entropy term); this issue was already noted by Blau & Michaeli (2018) as well.

## A.2  Using normalised LLM model scores

One potential solution to the problem we noted in the section above is to estimate the entropy term $\mathbb{H}[Q_{\mathbf{y}}^m]$.

**The ideal solution.** The cleanest solution would be to also estimate $\mathbb{H}[Q_{\mathbf{y}}^m]$ directly, and subtract it from our estimate. However, this requires us to have access the scores of the translations, i.e. we would want the dataset to comprise of $(\mathbf{y}_n^m, -\log q^m(\mathbf{y}_n))$ pairs, where $q^m(\mathbf{y}_n)$ is the probability mass function of distribution $Q^m$. Then, we could compute the perceptual score as:

$$D^m = \frac{1}{N} \sum_{n=1}^N \log \frac{q^m(\mathbf{y}_n)}{f(\mathbf{y}_n^m)}, \tag{14}$$

which would then converge to $D_{\mathrm{KL}}[Q_{\mathbf{y}}^m \| P_{\mathbf{y}}^{LM}]$ and would be comparable. Unfortunately, without the foresight to collect the model scores as well, this approach is not practical.

**Approximating the ideal solution.** An alternative, more feasible solution is to take inspiration from the outlier detection literature, and estimate the scores using a universal source coding algorithm such as Zip (Serrà et al., 2020). [7] Concretely, let $\ell^{zip}(\mathbf{y})$ denote the length of the Zip compressor's output in bits upon compressing the string $\mathbf{y}$. If $\mathbf{y}$ is long enough, $\ell^{zip}(\mathbf{y})$ should converge to $-\log q^m(\mathbf{y}_n)$, which warrants the estimate

$$D_{zip}^m = -\frac{1}{N} \sum_{n=1}^N (\log f(\mathbf{y}_n^m) + \ell^{zip}(\mathbf{y}_n^m)). \tag{15}$$

This approach can be justified by appealing to algorithmic information theory and in particular Kolmogorov complexity (Theis, 2024).

**A different interpretation.** These metrics admit an alternative interpretation as an approximation to a statistical distance which we describe in Section 4; see Appendix E for technical details.

# B   A generalisation of Theorem 1 of Blau & Michaeli

In the main text, we argued that for the purposes of translation the monolingual reference distribution $R_{\mathbf{y}}$ should not depend on $P_{\mathbf{x},\mathbf{y}}$, and derived the existence of the tradeoff from this principle.

In this section, we take a different approach and show that if we always set $R_{\mathbf{y}} \leftarrow P_{\mathbf{y}}$, our theory also admits a direct generalisation of Theorem 1 of Blau & Michaeli (2018), and hence a tradeoff exists in this case too. However, before we state the theorem, we make the following definitions.

**Definition B.1.** *A distortion measure $\Delta : \mathcal{X} \times \mathcal{Y} \times \mathcal{Y} \to \mathbb{R}$ is distribution preserving at $P_{\mathbf{x},\mathbf{y}}$ with respect to $P_{\mathbf{y}}$ if for the system $Q_{\mathbf{y}|\mathbf{x}}^*$ maximising Equation (1) we have $Q_{\mathbf{y}}^* = P_{\mathbf{y}}$.*

---

[7]Technically, should take the minimum across several compressors for better approximation (Appendix C in Serrà et al. (2020)).

Now, assume that we have a $\Delta$ that is distribution preserving at $P_{\mathbf{x},\mathbf{y}}$; how good is it in other scenarios? Could $\Delta$ be distribution preserving if we change $P_{\mathbf{x},\mathbf{y}}$ slightly? This motivates the generalisation of Definition 2 of Blau & Michaeli (2018):

**Definition B.2.** *A distortion measure $\Delta$ is stably distribution preserving at $P_{\mathbf{x},\mathbf{y}}$ if it is distribution preserving at all $\tilde{P}_{\mathbf{x},\mathbf{y}}$ in a TV $\epsilon$-ball around $P_{\mathbf{x},\mathbf{y}}$ for some $\epsilon > 0$.*

Now, we have the following generalisation of Theorem 1 of Blau & Michaeli (2018).

**Theorem B.1.** *If the relationship from $\mathbf{x}$ to $\mathbf{y}$ is not deterministic in the sense that $\mathbb{H}[\mathbf{y} \mid \mathbf{x}] > 0$, then $\Delta$ is not stably distribution preserving at $P_{\mathbf{x},\mathbf{y}}$ with respect to $R_{\mathbf{y}}(P_{\mathbf{y}})$.*

Our proof technique is a straight-forward extension of the Blau and Micali's proof. By way of contradiction, we assume that there is a $\Delta$ that is stably distribution preserving. Then: 1) we show that a the minimizer of Eq 3 is unique when $\Delta$ is stably distribution preserving and then 2) we show that the non-invertibility between $\mathbf{x}$ and $\mathbf{y}$ implies that the minimizer is non-unique. Taking these two facts together, we obtain a contradiction, hence the result holds.

Next, we define the following notion of non-uniqueness for the minimizer of a distortion measure $\Delta$:

**Definition B.3.** *Let $\Delta$ be a distortion metric (see Section 2). We say that the minimizer of $\Delta$ is non-unique if there exist two distributions $Q^1_{\mathbf{y}|\mathbf{x}}, Q^2_{\mathbf{y}|\mathbf{x}}$ such that*

$$\Delta(Q^1_{\mathbf{y}|\mathbf{x}}) = \Delta(Q^2_{\mathbf{y}|\mathbf{x}}) = \min_{Q_{\mathbf{y}|\mathbf{x}}} \Delta(Q_{\mathbf{y}|\mathbf{x}})$$

*and we have*

$$\mathbb{E}_{\mathbf{x} \sim P_{\mathbf{x}}}[D_{TV}(Q^1_{\mathbf{y}|\mathbf{x}}, Q^2_{\mathbf{y}|\mathbf{x}})] > 0.$$

The proof proceeds analogously to the proof of Theorem 1 in Blau & Michaeli (2018). We now proceed to prove Theorem B.1:

*Proof.* We first show, that our notion of non-determinism is equivalent to Blau & Michaeli's notion of "non-invertibility"

**Lemma B.1.** *Let $\mathbf{x}, \mathbf{y} \sim P_{\mathbf{x},\mathbf{y}}$ be random variables over $\mathcal{X} \times \mathcal{Y}$. Then, following are equivalent:*

1. ***Non-determinism:*** *$\mathbb{H}[\mathbf{y} \mid \mathbf{x}] > 0$.*

2. ***Non-invertibility:*** *There exists $S_{\mathbf{x}} \subseteq \mathcal{X}$ and $S_{\mathbf{y}} \subseteq \mathcal{Y}$ with $P_{\mathbf{x}}(S_{\mathbf{x}}) > 0$ and $|S_{\mathbf{y}}| > 1$ such that $P_{\mathbf{y}|\mathbf{x}}$ is either 1) supported on an uncountable set or 2) admits a probability mass function $p(y \mid x)$ and it holds that $\forall x, y \in S_{\mathbf{x}} \times S_{\mathbf{y}}$ we have $p(y \mid x) > 0$.*

*Proof.* The conditional entropy $\mathbb{H}[\mathbf{y} \mid \mathbf{x}] = 0$ when and only when $P_{\mathbf{y}|\mathbf{x}}$ admits a probability mass function $p(y \mid x)$ and there exists a function $g$ such that $P_{\mathbf{x}}$-almost surely $p(y \mid x) = \mathbf{1}[y = g(x)]$ (Exercise 2.5; Cover & Thomas, 2012). Then, both directions of the lemma follow directly from the contraposition of the above equivalence. □

We believe our definition is clearer than the equivalent definition of Blau & Michaeli (2018), as it makes intuitively more sense: the relationship $\mathbf{x} \to \mathbf{y}$ is non-deterministic, because knowing $\mathbf{x}$ still leaves some uncertainty in $\mathbf{y}$.

We proceed with the following uniqueness result.

**Lemma B.2.** *If $\Delta$ is a stably distribution preserving at $P_{\mathbf{x},\mathbf{y}}$ with respect to $R_{\mathbf{y}}(\cdot)$, then the minimizer $Q^*_{\mathbf{y}|\mathbf{x}} = \arg\min_{Q_{\mathbf{y}|\mathbf{x}}} \Delta(Q_{\mathbf{y}|\mathbf{x}})$ is uniquely defined by $P_{\mathbf{y}|\mathbf{x}}$.*

*Proof.* The proof of the lemma relies on the following two facts:

**Fact 1: The minimizer of $\Delta$ does not depend on the input distribution.** To see this, note that by definition,

$$\Delta(Q_{\mathbf{y}|\mathbf{x}}) = \mathbb{E}_{\mathbf{x},\mathbf{y}^r \sim P_{\mathbf{x},\mathbf{y}}}[\mathbb{E}_{\mathbf{y}^c \sim Q_{\mathbf{y}|\mathbf{x}}}[\Delta(\mathbf{x},\mathbf{y}^r,\mathbf{y}^c)]]$$
$$= \mathbb{E}_{\mathbf{x} \sim P_{\mathbf{x}}}\left[\mathbb{E}_{\mathbf{y}^r \sim P_{\mathbf{y}|\mathbf{x}},\mathbf{y}^c \sim Q_{\mathbf{y}|\mathbf{x}}}[\Delta(\mathbf{x},\mathbf{y}^r,\mathbf{y}^c)]\right]$$
$$= \mathbb{E}_{\mathbf{x} \sim P_{\mathbf{x}}}\left[f(\mathbf{x},Q_{\mathbf{y}|\mathbf{x}})\right],$$

where we defined

$$f(\mathbf{x},Q_{\mathbf{y}|\mathbf{x}}) = \mathbb{E}_{\mathbf{y}^r \sim P_{\mathbf{y}|\mathbf{x}},\mathbf{y}^c \sim Q_{\mathbf{y}|\mathbf{x}}}[\Delta(\mathbf{x},\mathbf{y}^r,\mathbf{y}^c)].$$

Now, we note the following "minimean" result (somewhat analogously to a minimax theorem):

$$\min_{Q_{\mathbf{y}|\mathbf{x}}} \Delta(Q_{\mathbf{y}|\mathbf{x}}) = \min_{Q_{\mathbf{y}|\mathbf{x}}} \mathbb{E}_{\mathbf{x} \sim P_{\mathbf{x}}}\left[f(\mathbf{x},Q_{\mathbf{y}|\mathbf{x}})\right] = \mathbb{E}_{\mathbf{x} \sim P_{\mathbf{x}}}\left[\min_{Q_{\mathbf{y}|\mathbf{x}}} f(\mathbf{x},Q_{\mathbf{y}|\mathbf{x}})\right]. \tag{16}$$

To see this, note that for all $Q_{\mathbf{y}|\mathbf{x}}$ it holds that

$$\mathbb{E}_{\mathbf{x} \sim P_{\mathbf{x}}}\left[f(\mathbf{x},Q_{\mathbf{y}|\mathbf{x}})\right] \geq \mathbb{E}_{\mathbf{x} \sim P_{\mathbf{x}}}\left[\min_{Q_{\mathbf{y}|\mathbf{x}}} f(\mathbf{x},Q_{\mathbf{y}|\mathbf{x}})\right]$$

However, setting $Q^*_{\mathbf{y}|\mathbf{x}=x} = \arg\min_{Q_{\mathbf{y}|\mathbf{x}=x}} f(x,Q_{\mathbf{y}|\mathbf{x}=x})$ for each $x$, we see that

$$\mathbb{E}_{\mathbf{x} \sim P_{\mathbf{x}}}[f(\mathbf{x},Q^*_{\mathbf{y}|\mathbf{x}})] = \mathbb{E}_{\mathbf{x} \sim P_{\mathbf{x}}}\left[\min_{Q_{\mathbf{y}|\mathbf{x}}} f(\mathbf{x},Q_{\mathbf{y}|\mathbf{x}})\right].$$

Now, since by Equation (16) the expectation with respect to $P_{\mathbf{x}}$ and the minimization are exchangeable, **the minimizer $Q^*_{\mathbf{y}|\mathbf{x}}$ does not depend on $P_{\mathbf{x}}$.**

**Fact 2: Convex perturbations are in the $\epsilon$-TV ball of $P_{\mathbf{x},\mathbf{y}}$.** Next, note that if $\Delta$ is stably distribution preserving at $P_{\mathbf{x},\mathbf{y}}$ within an $\epsilon > 0$ radius TV-ball, then it is distribution preserving at any joint distribution $\tilde{P}_{\mathbf{x},\mathbf{y}} = P_{\mathbf{y}|\mathbf{x}}\tilde{P}_{\mathbf{x}},$[8] where

$$\tilde{P}_{\mathbf{x}} = \alpha P_{\mathbf{x}} + (1-\alpha)Q_{\mathbf{x}}$$

where $\alpha \geq 1 - \epsilon$ and $Q_{\mathbf{x}}$ is any distribution over $\mathcal{X}$. This follows, since $\tilde{P}_{\mathbf{x},\mathbf{y}}$ is in the $\epsilon$-TV ball around $P_{\mathbf{x},\mathbf{y}}$:

$$D_{TV}(P_{\mathbf{x},\mathbf{y}},\tilde{P}_{\mathbf{x},\mathbf{y}}) = D_{TV}(P_{\mathbf{x},\mathbf{y}},\alpha P_{\mathbf{y}|\mathbf{x}}P_{\mathbf{x}} + (1-\alpha)P_{\mathbf{y}|\mathbf{x}}Q_{\mathbf{x}})$$
$$\leq \alpha D_{TV}(P_{\mathbf{x},\mathbf{y}},P_{\mathbf{y}|\mathbf{x}}P_{\mathbf{x}}) + (1-\alpha)D_{TV}(P_{\mathbf{x},\mathbf{y}},P_{\mathbf{y}|\mathbf{x}}Q_{\mathbf{x}}) \quad \text{(convexity of } D_{TV})$$
$$\leq (1-\alpha) \quad \text{(first term vanishes and } D_{TV} \leq 1.)$$
$$\leq \epsilon. \tag{17}$$

Let $\tilde{P}_{\mathbf{y}}$ denote the marginal distribution defined by the joint $\tilde{P}_{\mathbf{x},\mathbf{y}}$, given by

$$\tilde{P}_{\mathbf{y}}(\cdot) = \mathbb{E}_{\mathbf{x} \sim \tilde{P}_{\mathbf{x}}}[P_{\mathbf{y}|\mathbf{x}}(\cdot)]$$
$$= \alpha\mathbb{E}_{\mathbf{x} \sim P_{\mathbf{x}}}[P_{\mathbf{y}|\mathbf{x}}(\cdot)] + (1-\alpha)\mathbb{E}_{\mathbf{x} \sim Q_{\mathbf{x}}}[P_{\mathbf{y}|\mathbf{x}}(\cdot)]$$
$$= \alpha P_{\mathbf{y}} + (1-\alpha)\mathbb{E}_{\mathbf{x} \sim Q_{\mathbf{x}}}[P_{\mathbf{y}|\mathbf{x}}(\cdot)] \tag{18}$$

---

[8]For convenicence, let $P_{\mathbf{y}|\mathbf{x}}\tilde{P}_{\mathbf{x}}$ denote the semi-direct product of $P_{\mathbf{y}|\mathbf{x}}$ with $\tilde{P}_{\mathbf{x}}$, that is for $S \in \mathcal{X} \times \mathcal{Y}$ we have $P_{\mathbf{y}|\mathbf{x}}\tilde{P}_{\mathbf{x}}(S) = \mathbb{E}_{\mathbf{x} \sim \tilde{P}_{\mathbf{x}}}\mathbb{E}_{\mathbf{x} \sim P_{\mathbf{y}|\mathbf{x}}}[\mathbf{1}[(\mathbf{x},\mathbf{y}) \in S]].$

Furthermore, by Fact 1, since minimizer of $\Delta$ does not depend on the distribution of $\mathbf{x}$, we have that $\tilde{Q}^*_{\mathbf{y}|\mathbf{x}} = Q^*_{\mathbf{y}|\mathbf{x}}$. Letting $\tilde{P}_\mathbf{x} = \alpha P_\mathbf{x} + (1-\alpha)Q_\mathbf{x}$ be the $\mathbf{x}$-marginal of $\tilde{P}_{\mathbf{x},\mathbf{y}}$. This now allows us to compute $\tilde{Q}^*_\mathbf{y}$:

$$
\begin{aligned}
\tilde{Q}^*_\mathbf{y}(\cdot) &= \mathbb{E}_{\mathbf{x}\sim\tilde{P}_\mathbf{x}}[Q^*_{\mathbf{y}|\mathbf{x}}(\cdot)] \\
&= \alpha\mathbb{E}_{\mathbf{x}\sim P_\mathbf{x}}[Q^*_{\mathbf{y}|\mathbf{x}}(\cdot)] + (1-\alpha)\mathbb{E}_{\mathbf{x}\sim Q_\mathbf{x}}[Q^*_{\mathbf{y}|\mathbf{x}}(\cdot)] \\
&= \alpha Q^*_\mathbf{y}(\cdot) + (1-\alpha)\mathbb{E}_{\mathbf{x}\sim Q_\mathbf{x}}[Q^*_{\mathbf{y}|\mathbf{x}}(\cdot)] \\
&= \alpha P_\mathbf{y} + (1-\alpha)\mathbb{E}_{\mathbf{x}\sim Q_\mathbf{x}}[Q^*_{\mathbf{y}|\mathbf{x}}(\cdot)]
\end{aligned}
\tag{19}
$$

Now, since $\Delta$ is distribution preserving at $\tilde{P}_{\mathbf{x},\mathbf{y}}$ by Fact 2, we must have $\tilde{Q}^*_\mathbf{y} = \tilde{P}_\mathbf{y}$, where $\tilde{Q}^*_\mathbf{y}$ is the marginal distribution defined by the minimizer $\tilde{Q}^*_{\mathbf{y}|\mathbf{x}}$ of $\Delta$ with respect to $\tilde{P}_{\mathbf{x},\mathbf{y}}$. This means that Equations (18) and (19) must be equal, hence we must also have

$$
\mathbb{E}_{\mathbf{x}\sim Q_\mathbf{x}}[Q^*_{\mathbf{y}|\mathbf{x}}(\cdot)] = \mathbb{E}_{\mathbf{x}\sim Q_\mathbf{x}}[P_{\mathbf{y}|\mathbf{x}}(\cdot)].
$$

Since $Q_\mathbf{x}$, was arbitrary, this equation holds if and only if $Q^*_{\mathbf{y}|\mathbf{x}} = P_{\mathbf{y}|\mathbf{x}}$ as required. $\qquad\square$

We now move onto our second result:

**Lemma B.3.** *Assume the relation from $\mathbf{x} \to \mathbf{y}$ is non-deterministic. Furthermore, assume that the minimizer of $\Delta$ is $Q^*_{\mathbf{y}|\mathbf{x}} = P_{\mathbf{x}|\mathbf{y}}$. Then, $Q^*_{\mathbf{y}|\mathbf{x}}$ is non-unique.*

*Proof.* Follows from Lemma B.1 combined with Lemma 2 from Blau & Michaeli (2018). $\qquad\square$

To finish the proof, assume by way of contradiction that the $P_{\mathbf{x},\mathbf{y}}$ defines a non-deterministic relationship from $\mathbf{x} \to \mathbf{y}$ and that the distortion $\Delta$ is stably distribution preserving at $P_{\mathbf{x},\mathbf{y}}$. Then, by Lemma B.2, the minimizer of $\Delta$ is $Q^*_{\mathbf{y}|\mathbf{x}} = P_{\mathbf{y}|\mathbf{x}}$ and this estimator is unique. However, by Lemma B.3, $Q^*_{\mathbf{y}|\mathbf{x}}$ is non-unique, which leads to a contradiction. $\qquad\square$

## C  Proof of Theorem 3.1

**Theorem 3.1.** *Let $A(N)$ be the accuracy-naturalness function defined by distortion $\Delta$, distributional distance $D$, parallel distribution $P_{\mathbf{x},\mathbf{y}}$ and monolingual reference $R_\mathbf{y}$. Then:*

1. *$A(N)$ is **non-increasing** in N.*

2. *if $D$ is convex in its first slot, then $A(N)$ is **concave** in N.*

*Proof.* **(1) Non-increasing.** We study the set of admissible translation systems. Thus, let

$$
C_N = \{Q_{\mathbf{y}|\mathbf{x}} \mid -D(Q_\mathbf{y}, R_\mathbf{y}) \geq N\}.
\tag{20}
$$

Now note that since $-D$ is bounded from above, for $N \geq N'$ we have $-D(Q_\mathbf{y}, R_\mathbf{y}) \geq N \geq N'$, hence $C_N \subseteq C_{N'}$. Therefore, $A(N)$ is the supremum of the same objective but over a bigger constraint set, hence we must have $A(N) \leq A(N')$.

**(2) Concave when $D$ is convex in the first slot.** We wish to prove for all $N_0, N_1$ in the range of $D$ and $\lambda \in [0,1]$ that

$$
A(\lambda N_0 + (1-\lambda)N_1) \geq \lambda A(N_0) + (1-\lambda)A(N_1).
\tag{21}
$$

To begin, let

$$
Q^0_{\mathbf{y}|\mathbf{x}} = \arg\max_{Q_{\mathbf{y}|\mathbf{x}}} -\Delta(Q_{\mathbf{y}|\mathbf{x}}) \quad \text{s.t.} \ -D(Q_\mathbf{y}, R_\mathbf{y}) \geq N_0
\tag{22}
$$

$$
Q^1_{\mathbf{y}|\mathbf{x}} = \arg\max_{Q_{\mathbf{y}|\mathbf{x}}} -\Delta(Q_{\mathbf{y}|\mathbf{x}}) \quad \text{s.t.} \ -D(Q_\mathbf{y}, R_\mathbf{y}) \geq N_1.
\tag{23}
$$

In other words, $-\Delta(Q^0_{\mathbf{y}|\mathbf{x}}) = A(N_0)$ and $-\Delta(Q^1_{\mathbf{y}|\mathbf{x}}) = A(N_1)$. This allows us to rewrite Equation (21) as

$$A(\lambda N_0 + (1 - \lambda)N_1) \geq -\lambda \Delta(Q^0_{\mathbf{y}|\mathbf{x}}) - (1 - \lambda)\Delta(Q^1_{\mathbf{y}|\mathbf{x}}) \tag{24}$$

Next, we define

$$Q^\lambda_{\mathbf{y}|\mathbf{x}} = \lambda Q^0_{\mathbf{y}|\mathbf{x}} + (1 - \lambda)Q^1_{\mathbf{y}|\mathbf{x}}.$$

Now, let $Q^\lambda_{\mathbf{y}}(\cdot) = \mathbb{E}_{\mathbf{x} \sim P_{\mathbf{x}}}[Q^\lambda_{\mathbf{y}|\mathbf{x}}(\cdot)]$, and define $Q^0_{\mathbf{y}}$ and $Q^1_{\mathbf{y}}$ analogously. Furthermore, let $N_\lambda = -D(Q^\lambda_{\mathbf{y}}, R_{\mathbf{y}})$. Then, since $D$ is convex in its first argument by assumption, we have that

$$\begin{aligned}
N_\lambda &= -D(Q^\lambda_{\mathbf{y}}, R_{\mathbf{y}}) \\
&\geq -\lambda D(Q^0_{\mathbf{y}}, R_{\mathbf{y}}) - (1 - \lambda)D(Q^1_{\mathbf{y}}, R_{\mathbf{y}}) \\
&\geq \lambda N_0 + (1 - \lambda)N_1 \qquad \text{(by the constraints in eqs. (22) and (23))}
\end{aligned}$$

Combining this with the fact that $A(N)$ is non-increasing in $N$, we find that

$$A(\lambda N_0 + (1 - \lambda)N_1) \geq A(N_\lambda).$$

Hence, to show Equation (24), it remains to show that

$$A(N_\lambda) \geq -\lambda \Delta(Q^0_{\mathbf{y}|\mathbf{x}}) - (1 - \lambda)\Delta(Q^1_{\mathbf{y}|\mathbf{x}}).$$

To see this, observe that

$$\begin{aligned}
A(N_\lambda) &= \max_{Q_{\mathbf{y}|\mathbf{x}}} -\Delta(Q_{\mathbf{y}|\mathbf{x}}) \quad \text{s.t.} \; -D(Q_{\mathbf{y}}, R_{\mathbf{y}}) \geq N_\lambda \\
&\geq -\Delta(Q^\lambda_{\mathbf{y}|\mathbf{x}}),
\end{aligned}$$

where the inequality follows since $-D(Q^\lambda_{\mathbf{y}}, R_{\mathbf{y}}) = N_\lambda$ by definition, hence $-D(Q_{\mathbf{y}}, R_{\mathbf{y}}) \geq N_\lambda$ is in the admissible set. Finally, note that

$$\begin{aligned}
-\Delta(Q^\lambda_{\mathbf{y}|\mathbf{x}}) &= -\mathbb{E}_{\mathbf{x},\mathbf{y} \sim P_{\mathbf{x},\mathbf{y}}}\mathbb{E}_{\mathbf{y}' \sim Q^\lambda_{\mathbf{y}|\mathbf{x}}}[\Delta(\mathbf{x}, \mathbf{y}, \mathbf{y}')] \tag{25} \\
&= -\mathbb{E}_{\mathbf{x},\mathbf{y} \sim P_{\mathbf{x},\mathbf{y}}}[\lambda \mathbb{E}_{\mathbf{y}' \sim Q^0_{\mathbf{y}|\mathbf{x}}}[\Delta(\mathbf{x}, \mathbf{y}, \mathbf{y}')] + (1 - \lambda)\mathbb{E}_{\mathbf{y}' \sim Q^0_{\mathbf{y}|\mathbf{x}}}[\Delta(\mathbf{x}, \mathbf{y}, \mathbf{y}')]] \quad \text{(definition)} \\
&= -\lambda\Delta(Q^0_{\mathbf{y}|\mathbf{x}}) - (1 - \lambda)\Delta(Q^1_{\mathbf{y}|\mathbf{x}}), \quad \text{(definition)}
\end{aligned}$$

which finishes the proof. $\qquad\square$

## D   MQM details

For the en $\to$ de and ja $\to$ zh language pairs, the WMT24 evaluation exactly followed the evaluation scheme proposed in Freitag et al. (2021). The error categories are shown in Table 1 (limited to the actual errors found in the data), together with our classification into accuracy or fluency errors. This classification is straightforward, as most categories are already labelled as such. It is worth noting that we ignored the "Source issue" and "Other" categories, as we found the marked errors to be uninformative in most cases. In principle it would be possible to classify most of the individual errors in this category to either adequacy or fluency, but it would involve a huge amount of additional manual work. The en $\to$ es translation used another error schema. Its classification into adequacy and fluency errors is shown in Table 2.

Once we have the classification into these categories we can compute adequacy and fluency scores using the same weighting schema for errors as given in Table 4 in Freitag et al. (2021).

| Accuracy Errors | Fluency Errors | Other |
|---|---|---|
| Accuracy/Addition
Accuracy/Creative Reinterpretation
Accuracy/Gender Mismatch
Accuracy/Mistranslation
Accuracy/Omission
Accuracy/Source language fragment
Non-translation! | Fluency/Grammar
Fluency/Inconsistency
Fluency/Punctuation
Fluency/Register
Fluency/Spelling
Fluency/Text-Breaking
Locale convention/Address format
Locale convention/Currency format
Locale convention/Time format
Style/Archaic or obscure word choice
Style/Bad sentence structure
Style/Unnatural or awkward
Terminology/Inappropriate for context
Terminology/Inconsistent | Other
Source issue |

Table 1: Our classification of MQM errors into adequacy and fluency errors for the en → de and ja → zh translation directions.

| Accuracy Errors | Fluency Errors | Other |
|---|---|---|
| Addition
Agreement
Do not translate
Mistranslation
MT hallucination
Omission
Untranslated
Wrong named entity
Wrong term | Capitalization
Date-time format
Inconsistency
Lacks creativity
Grammar
Measurement format
Number format
Punctuation
Register
Spelling
Unnatural flow
Whitespace
Word order
Wrong language variety | Other
Source issue |

Table 2: Our classification of MQM errors into adequacy and fluency errors for the en → es translation direction.

# E    Technical details for the divergence in Section 4.1

In this section, we define a generalised variant of the divergence from Section 4.1. Our starting point is to notice that IPMs could be viewed as a certain infinity-norm, since we taking a supremum as part of their definition. This suggests that we should be able to define analogous $p$-norms for $p \in [1, \infty)$ over appropriate spaces of distributions too.

We base the following discussion on Müller (1997) to make the above idea precise. Let $S$ be a measurable space and $b : S \to [1, \infty)$ a measurable function. Let $\mathfrak{B}_b$ be the set of measurable functions $f : S \to \mathbb{R}$ such that

$$\|f\|_b = \sup_{s \in S} \frac{|f(s)|}{b(s)} < \infty. \tag{26}$$

For example, taking $b = 1$ we get that $\mathfrak{B}_1$ is the set of bounded functions from $S$ to $\mathbb{R}$. Next, let $\mathbb{M}_b$ be the set of all signed measures $\mu$ such that $|\mu|(b) = \mu^+(b) + \mu^-(b) < \infty$, and let $\mathbb{P}$ denote the set of all probability metrics over $S$. Then, by Lemma 2.1 of Müller (1997), $\mathbb{M}_b$ and $\mathfrak{B}_b$ are in strict duality under the dual pairing

$$\langle \cdot, \cdot \rangle : \mathbb{M}_b \times \mathfrak{B}_b \to \mathbb{R}$$
$$\langle \mu, f \rangle = \mu(f).$$

Next, define $\mathbb{M}_b^N \subset \mathbb{M}_b$ as the set of all signed measures $\mu$ with $\mu(S) = 0$, and $\mathfrak{B}_b^\sim$ as the quotient space of $\mathfrak{B}_b$ by the equivalence relation $f \sim g \Leftrightarrow f - g$ is constant. Then, Lemma 2.2 of Müller (1997) shows that $\mathbb{M}_b^N$ and $\mathfrak{B}_b^\sim$ are also in strict duality under the same bilinear map as above. Now, for a function class $\mathcal{F} \subseteq \mathfrak{B}_b$, we can define the following seminorm on $\mathbb{M}_b^N$:

$$\|\mu\|_{\mathcal{F}} = \sup_{f \in \mathcal{F}} |\mu(f)|. \tag{27}$$

Importantly, this seminorm induces the *integral probability metric* $d_{\mathcal{F}}$ over $\mathbb{P}$:

$$d_{\mathcal{F}}(Q, P) = \|Q - P\|_{\mathcal{F}}. \tag{28}$$

Note that $\|\cdot\|_{\mathcal{F}}$ is the $\infty$-seminorm on $\mathbb{M}_b^N$. This motivates us to define a larger family of seminorms as follows. Let $\mathcal{P}$ be a probability measure over $\mathfrak{B}_b$. Now, define the seminorm

$$\|\mu\|_{p,\mathcal{P}} = \mathcal{P}(|\mu|^p)^{1/p} \tag{29}$$
$$= \mathbb{E}_{f \sim \mathcal{P}}[|\mu(f)|^p]^{1/p} \tag{30}$$

Then, similarly to $\|\cdot\|_{\mathcal{F}}$, this induces a semimetric on $\mathbb{P}$:

$$D_p(Q, P \mid \mathcal{P}) = \|Q - P\|_{p,\mathcal{P}} = \mathcal{P}(|Q - P|^p)^{1/p} \tag{31}$$

Now that we defined our divergence, we show that it admits a similar connection to classification as IPMs, analogously to Theorem 17 of Sriperumbudur et al. (2009):

**Theorem E.1** (Connection to expected classification risk). *Let $\mathcal{F}, P, Q, D_p$ for $p \in [0, \infty)$ be as above. Then: For a loss function $L : -1, 1 \times \mathbb{R} \to \mathbb{R}$, define the expected binary classification risk of $\mathcal{P}$ as*

$$R_{\mathcal{P}}^L = \mathbb{E}_{f \sim \mathcal{P}}[\mathbb{E}_{X,Y}[L_Y(f(X))]] \tag{32}$$
$$= \mathbb{E}_{f \sim \mathcal{P}}[\epsilon \mathbb{E}_{X \sim P}[L_1(f(x))] + (1 - \epsilon)\mathbb{E}_{X \sim Q}[L_{-1}(f(x))]], \tag{33}$$

*where $\epsilon = \mathbb{P}[Y = 1]$, $P = \mathbb{P}[X \mid Y = 1]$ and $Q = \mathbb{P}[X \mid Y = -1]$. Then, for $L_1(\alpha) = -\alpha/\epsilon$ and $L_{-1}(\alpha) = \alpha/(1 - \epsilon)$, we have*

$$R_\infty^L = -D_\infty(P, Q \mid \mathcal{P}) \leq -D_1(P, Q \mid \mathcal{P}) \leq R_{\mathcal{P}}^L \tag{34}$$

*Proof.* Define $(x)_+ = \max\{0, x\}$, and $k(f) = \mathbb{E}[f(X) \mid Y = -1] - \mathbb{E}[f(X) \mid Y = 1]$. Then, with our choice for the loss function, we have

$$
\begin{aligned}
R_{\mathcal{P}}^L &= \mathbb{E}_f \left[ \mathbb{E}[f(X) \mid Y = -1, f] - \mathbb{E}[f(X) \mid Y = 1, f] \right] && \text{(def)} \\
&= \mathbb{E}_f \left[ k(f) \right] && (f \perp X, Y) \\
&= -\mathbb{E}_f \left[ |k(f)| \right] + 2 \cdot \mathbb{E}_f \left[ (k(f))_+ \right] && (x = -|x| + 2(x)_+) \\
&\geq -D_1(Q, P \mid \mathcal{P}) && \text{(def)} \\
&= -\mathbb{E}_f \left[ \left( |k(f)|^p \right)^{1/p} \right] \\
&\geq -\mathbb{E}_f \left[ |k(f)|^p \right]^{1/p} && \text{(Jensen for } p \geq 1) \\
&= -D_p(Q, P \mid \mathcal{P}) && \text{(def)} \\
&\to -D_\infty(Q, P \mid \mathcal{P}) \quad \text{as } p \to \infty \\
&= R_\infty^L && \text{(Theorem 17 of Sriperumbudur et al. (2009))}
\end{aligned}
$$

$\square$

## E.1   Examples of $D_p$ when $\mathcal{P}$ is of Gaussian process type

Based on its definition, $D_p$ can be easily approximated via a Monte Carlo estimate for both samples from the process $\mathcal{P}$ and samples from $Q$ and $P$. However, in this section, we show that for a certain broad class of processes, we can evaluate the expectation over samples of $\mathcal{P}$. More precisely, we compute $D_p$ when $\mathcal{P}$ is of *Gaussian process type*, i.e., for $f \sim \mathcal{P}$ we have

$$(\lambda \circ f) \sim \mathcal{GP}(m, k)$$

for link function $\lambda$ and Gaussian process $\mathcal{GP}(m, k)$ with mean function $m$ and covariance function $k$ and where $\circ$ indicates function composition. It turns out, that in these cases, $D_p$ is intimately related to the maximum mean discrepancy $\text{MMD}_k$ (Gretton et al., 2012), where the MMD is with respect to a reproducing kernel Hilbert space with the kernel $k$ taken to be the covariance function of the Gaussian process above. Indeed, we show in Appendix E.1.1 that in certain cases, $D_p$ and $\text{MMD}_k$ are even identical up to a multiplicative constant. However, as we are averaging instead of maximising in the case of $D_p$, in such cases we might call it *mean-mean discrepancy* (MeMeD).

Now, we have the following result for the MeMeD when $p = 2$:

**Theorem E.2.** *Let $P$ and $Q$ be probability measures over $\mathcal{X}$. Let $\mathcal{P}$ be of Gaussian process type with mean function $m$, covariance function $k$ and link function $\lambda$, i.e. for $f \sim \mathcal{P}$ we have $(\lambda \circ f) \sim \mathcal{GP}(m, k)$. Then, setting*

$$C(x, y) = \mathbb{E}_{f \sim \mathcal{P}}[f(x)f(y)] = \mathbb{E}_{g \sim \mathcal{GP}(m,k)}[\lambda^{-1}(g(x))\lambda^{-1}(g(y))] \tag{35}$$

*we get that*

$$D_2(Q, P \mid \mathcal{P})^2 = \left\langle Q - P, \int C(x, y) \, d(Q - P)(y) \right\rangle_{\mathbb{M}_b} \tag{36}$$

$$= \mathbb{E}_{X,Y \sim Q^{\otimes 2}}[C(X, Y)] - 2\mathbb{E}_{X,Y \sim Q \otimes P}[C(X, Y)] + \mathbb{E}_{X,Y \sim P^{\otimes 2}}[C(X, Y)]. \tag{37}$$

*Furthermore, for a dataset of $N$ samples $\{X_n\}_{n=1}^N$ of $Q$ and $M$ samples $\{Y_m\}_{m=1}^M$ of $P$, then $D_2$ admits the following unbiased estimator:*

$$D_2(Q, P \mid \mathcal{P})^2 \approx \frac{1}{N(N-1)} \sum_{\substack{1 \leq i,j \leq N \\ i \neq j}} C(X_i, X_j) + \frac{1}{M(M-1)} \sum_{\substack{1 \leq i,j \leq M \\ i \neq j}} C(Y_i, Y_j)$$

$$- \frac{2}{NM} \sum_{\substack{1 \leq i \leq N \\ 1 \leq j \leq M}} C(X_i, Y_j) \tag{38}$$

*Proof.* By definition, we have

$$D_2(Q, P \mid \mathcal{P})^2 = \mathbb{E}_{f \sim \mathcal{P}}[\langle Q - P, f \rangle^2] \tag{39}$$

$$= \mathbb{E}_{f \sim \mathcal{P}}[\langle Q - P, T_f(Q - P) \rangle_{\mathbb{M}_b}] \tag{40}$$

where the first set of angle brackets denotes the dual pairing as above, while the second set of angle brackets denote the inner product over $\mathbb{M}_b$. Furthermore, $T_f$ is the integral operator defined by

$$\pi \in \mathbb{M}_b : \quad (T_f \pi)(x) = \int f(x) f(y) \, d\pi(y). \tag{41}$$

Then, by the linearity of expectation, we get

$$D_2(Q, P \mid \mathcal{P})^2 = \langle Q - P, \mathbb{E}_{f \sim \mathcal{P}}[T_f](Q - P) \rangle_{\mathbb{M}_b}. \tag{42}$$

To compute $\mathbb{E}_{f \sim \mathcal{P}}[T_f]$, note that by the law of the unconscious statistician, we have for any $\pi \in \mathbb{M}_b$

$$(\mathbb{E}_{f \sim \mathcal{P}}[T_f] \pi)(x) = \int \mathbb{E}_{f \sim \mathcal{P}}[f(x) f(y)] \, d\pi(y) \tag{43}$$

$$= \int C(x, y) \, d\pi(y). \tag{44}$$

Substituting this back, proves the first part of our claim. Now, the fact Equation (38) is an unbiased estimator follows from the theory of U-statistics, similarly to the argument of Lemma 6 of Gretton et al. (2012). □

To demonstrate the power of Theorem E.2, we now consider three concrete cases when $\mathcal{P}$ is of Gaussian process type.

### E.1.1 When $\mathcal{P}$ is a Gaussian process

The simplest case of $\mathcal{P}$ being of Gaussian process type is when the link function is simply the identity. Indeed, in this case we can compute the entire family of $D_p$ metrics for all $p \in [0, \infty)$ without invoking Theorem E.2:

**Theorem E.3** (*D* based using Gaussian processes). *Let P and Q be probability measures over $\mathcal{X}$. Let $\mathcal{P} = \mathcal{GP}(m, k)$ be a Gaussian process with mean function m and kernel k over the sample space. Let $\Gamma(z) = \int_0^\infty t^{z-1} e^{-t} \, dt$ denote the Euler gamma function and $M(a, b, z)$ denote Kummer's confluent hypergeometric function. Then,*

$$D_p(P, Q \mid \mathcal{P})^p = \mathrm{MMD}_k(P, Q)^p \cdot 2^{p/2} \cdot \frac{\Gamma\left(\frac{1+p}{2}\right)}{\sqrt{\pi}} M\left(-\frac{p}{2}, \frac{1}{2}, -\frac{1}{2} \left(\frac{P(m) - Q(m)}{\mathrm{MMD}_k(P, Q)}\right)^2\right). \tag{45}$$

*In particular, we have*

$$D_2(P, Q \mid \mathcal{P})^2 = \mathrm{MMD}_k(P, Q)^2 + (P(m) - Q(m))^2$$

$$D_1(P, Q \mid \mathcal{P}) = \sqrt{\frac{2}{\pi}} \cdot \mathrm{MMD}_k(P, Q) \cdot \exp\left(\frac{-(P(m) - Q(m))^2}{2 \cdot \mathrm{MMD}_k(P, Q)^2}\right)$$

$$+ (P(m) - Q(m)) \left(1 - \Phi\left(\frac{P(m) - Q(m)}{\mathrm{MMD}_k(P, Q)}\right)\right). \tag{46}$$

*Furthermore, when $m = 0$, we have*

$$\sqrt{\frac{e}{p+1}} \cdot D_p(Q, P \mid \mathcal{P}) \to \mathrm{MMD}_k(P, Q) \quad \text{as } p \to \infty. \tag{47}$$

*Finally, the above results imply that $D_p$ metrizes weak convergence if and only if $\mathrm{MMD}_k$ does.*

*Proof.* By the consistency property of Gaussian processes, it holds that dual pairings of GPs with linear functionals are Gaussian distributed. Thus, for $f \sim \mathcal{GP}(m,k)$ we have

$$P(f) - Q(f) = \langle f, P - Q \rangle \sim \mathcal{N}(\langle m, P - Q \rangle, \langle P - Q, P - Q \rangle_k), \tag{48}$$

where

$$\langle P - Q, P - Q \rangle_k = \int \int k(x,y) \, d(P - Q)(x) \, d(P - Q)(y) = \text{MMD}_k(P,Q). \tag{49}$$

Now, set $\mu = \langle m, P - Q \rangle$ and $\sigma = \text{MMD}_k(P,Q)$. Then letting $\epsilon \sim \mathcal{N}(0,1)$, we have

$$D_p(P, Q \mid \mathcal{P})^p = \mathbb{E}[|\sigma\epsilon + \mu|^p]. \tag{50}$$

Then, applying equation 17 of Winkelbauer (2012) to the right-hand side of the above, we get

$$D_p(P, Q \mid \mathcal{P})^p = \sigma^p \cdot 2^{p/2} \cdot \frac{\Gamma\left(\frac{1+p}{2}\right)}{\sqrt{\pi}} M\left(-\frac{p}{2}, \frac{1}{2}, -\frac{1}{2}\left(\frac{\mu}{\sigma}\right)^2\right), \tag{51}$$

from which the identities for the distances hold. For the limit result, note that when $m = 0$, we have

$$D_p(P, Q \mid \mathcal{P}) = \sigma \cdot \sqrt{\frac{2}{\pi}} \cdot \Gamma\left(\frac{1+p}{2}\right)^{1/p} \tag{52}$$

The result then follows by applying Stirling's approximation. $\qquad\square$

### E.1.2 When $\mathcal{P}$ is a log-Gaussian process

An example that is perhaps closer to practice is requiring that the scores be non-negative. One way of achieving this is to set $\mathcal{P}$ to be a *log-Gaussian process*, that is, for $f \sim \mathcal{P}$, we have that $\log f \sim \mathcal{GP}(m, k)$ for some mean function $m$ and covariance function $k$.

**Theorem E.4.** *Let $P$ and $Q$ be probability measures over $\mathcal{X}$. Let $\mathcal{P}$ be a log-Gaussian process with mean function $m$ and covariance function $k$, i.e. for $f \sim \mathcal{P}$ we have $\log f \sim \mathcal{GP}(m, k)$. Then, $D_2$ can be computed according to Theorem E.2 by setting:*

$$\mu(x) = \mathbb{E}_{g \sim \mathcal{GP}(m,k)}[\exp(g(x))] = \exp\left(m(x) + \frac{k(x,x)}{2}\right), \tag{53}$$

$$C(x,y) = \mu(x)\mu(y) \exp(k(x,y)). \tag{54}$$

*Proof.* By applying Theorem E.2 with $\lambda = \log$, we simply need to compute

$$C(x,y) = \mathbb{E}_{f \sim \mathcal{P}}[f(x)f(y)] \tag{55}$$
$$= \mathbb{E}_{g \sim \mathcal{GP}(m,k)}[\exp(g(x) + g(y))] \tag{56}$$

Now, $g(x)$ and $g(y)$ are correlated Gaussian random variables, hence evaluating the expectation pointwise using the formulae from Halliwell (2015), we get

$$\mathbb{E}_{g \sim \mathcal{GP}(m,k)}[\exp(g(x) + g(y))] = \mu(x)\mu(y) \exp(k(x,y)) \tag{57}$$

$$\square$$

One interesting case of this distance is when we set $m(x) = -\frac{1}{|x|} \log p_{LM}(x)$ the negative log probability of a language model and set $k(x,y) = \mathbf{1}[x = y]$ (i.e. the scores constitute a pure noise process). Then, we see that $D_2$ becomes

$$D_2(Q, P \mid \mathcal{P}) = e \cdot \left| \mathbb{E}_{X \sim Q}\left[p_{LM}(X)^{-1/|x|}\right] - \mathbb{E}_{X \sim P}\left[p_{LM}(X)^{-1/|x|}\right] \right| \tag{58}$$

### *E.1.3 When $\mathcal{P}$ is a Probit process*

An example that is even closer to practice is requiring that the scores be bounded. One way of achieving this is to set $\mathcal{P}$ to be a *Probit process*. That is, letting $\Phi$ be the CDF of the standard Gaussian, for $f \sim \mathcal{P}$ we have that $\Phi^{-1}(f) \sim \mathcal{GP}(m,k)$ for some mean function $m$ and covariance function $k$. Then, we have the following result:

**Theorem E.5.** *Let $P$ and $Q$ be probability measures over $\mathcal{X}$. Let $\mathcal{P}$ be a Probit process with mean function $m$ and covariance function $k$, i.e. for $f \sim \mathcal{P}$ we have $\Phi^{-1}f \sim \mathcal{GP}(m,k)$. Then, $D_2$ can be computed according to Theorem E.2 by letting*

$$\mathcal{BN}(a,b \mid \rho) = \mathbb{P}_{[X,Y] \sim \mathcal{N}(0,\Sigma)}[X \le a, Y \le b] \qquad \text{where } \Sigma = \begin{bmatrix} 1 & \rho \\ \rho & 1 \end{bmatrix} \tag{59}$$

*denote the CDF of the standard bivariate normal with correlation $\rho$ evaluated at $(a,b)$, and setting*

$$C(x,y) = \mathcal{BN}\left(\frac{m(x)}{\sqrt{1+k(x,x)}}, \frac{m(y)}{\sqrt{1+k(y,y)}} \ \middle| \ \frac{k(x,y)}{\sqrt{1+k(x,x)}\sqrt{1+k(y,y)}}\right). \tag{60}$$

*Proof.* Similarly as in the previous section, we can apply Theorem E.2 with $\lambda = \Phi$. Then, we are interested in computing

$$\mathbb{E}_{f \sim \mathcal{P}}[f(x)f(y)] = \mathbb{E}_{g \sim \mathcal{GP}(m,k)}[\Phi(g(x))\Phi(g(y))] \tag{61}$$

$$= \mathbb{E}_{U,V \sim \mathcal{N}(\mu,\Sigma)}[\Phi(U)\Phi(V)] \tag{62}$$

$$\text{where } \mu = [m(x), m(y)] \text{ and } \Sigma = \begin{bmatrix} k(x,x) & k(x,y) \\ k(x,y) & k(y,y) \end{bmatrix} \tag{63}$$

Note, that $[U,V] = \mu + \Sigma^{1/2}\epsilon$, where $\epsilon \sim \mathcal{N}(0,I)$, from which we get $U = a\epsilon_1 + \mu_1$ and $V = b\epsilon_1 + c\epsilon_2 + \mu_2$, where $a = \sqrt{\Sigma_{11}}, b = \Sigma_{21}/a$ and $c = \sqrt{\Sigma_{22} - b^2}$ follow from the Cholesky decomposition of $\Sigma$. Hence, we find

$$\mathbb{E}_{U,V \sim \mathcal{N}(\mu,\Sigma)}[\Phi(U)\Phi(V)] = \mathbb{E}_{\epsilon \sim \mathcal{N}(0,I)}[\Phi(a\epsilon_1 + \mu_1)\Phi(b\epsilon_1 + c\epsilon_2 + \mu_2)]$$

$$= \mathbb{E}_{\epsilon_1 \sim \mathcal{N}(0,1)}[\Phi(a\epsilon_1 + \mu_1)\mathbb{E}_{\epsilon_2 \sim \mathcal{N}(0,1)}[\Phi(b\epsilon_1 + c\epsilon_2 + \mu_2) \mid \epsilon_1]]$$

$$= \mathbb{E}_{\epsilon_1 \sim \mathcal{N}(0,1)}\left[\Phi(a\epsilon_1 + \mu_1)\Phi\left(\frac{b\epsilon_1 + \mu_2}{\sqrt{1+c^2}}\right)\right]$$
$$\text{(identity 10,010.8 of Owen (1980))}$$

$$= \mathcal{BN}\left(\frac{\mu_1}{\sqrt{1+\Sigma_{11}}}, \frac{\mu_2}{\sqrt{1+\Sigma_{22}}} \ \middle| \ \frac{\Sigma_{21}}{\sqrt{1+\Sigma_{11}}\sqrt{1+\Sigma_{22}}}\right)$$
$$\text{(identity 20,010.3 of Owen (1980) \& simplifying)}$$

which finishes the proof. $\qquad\square$

## F Caveat with the LLM-based approximation of the AN curve

Note, that the approach we use to approximate the accuracy-naturalness curve in Section 5.1 will always produce an overly optimistic estimate for the curve, since we compute our estimate as

$$\mathcal{L}^* = \mathbb{E}[\max \mathcal{L}],$$

where $\mathcal{L}$ is the one-shot Lagrange dual of the AN function from Equation (5), and we maximise over systems/LLM-generated candidates for each translation first, and then we average over the different entries of the test set second. However, in the true Lagrange-dual of the AN function, the order of the expectation and the maximisation is swapped: $\max \mathbb{E}[\mathcal{L}]$. Finally, we have the standard result that

$$\mathbb{E}[\max \mathcal{L}] \ge \max \mathbb{E}[\mathcal{L}],$$

hence our procedure always overestimates the AN curve, i.e., it makes the achievable region look larger than it actually is.

