# OpenReview forum: "You Cannot Feed Two Birds with One Score: the Accuracy-Naturalness Tradeoff in Translation"
_colmweb.org/COLM/2025/Conference — COLM 2025_

### Official Review · Reviewer_z3nj · 2025-04-27

**Rating:** 9
**Confidence:** 5
**Ethics Flag:** 1

**Summary:**

This is a nice paper. It argues convincingly that there is a tradeoff between naturalness and accuracy when systems/people translate text. For example, when translating the German sentence "My teacher_FEMALE is nice" into English, the translation "My teacher is nice" is natural but only partially accurate because it does not make explicit that my teacher is female, which in German is represented explicitly, while the translation"My female teacher is nice" is un-natural - monolingual English texts are not written in this fashion.

The authors extend Blau and Michaeli's (2018) distorsion-perception theory to translation and show mathematically and empirically that there is a tradeoff between accuracy and naturalness when translating texts. Their results can be used to develop better evaluation techniques and enable translation systems to be tuned for different operational points in the accuracy/naturalness plane.

**Reasons To Accept:**

The paper introduces a nice theoretical/practical framework for studying the translation process and evaluating translations produced by humans and/or machines in a richer two dimensional space defined by the accuracy and naturalness axes.

The authors develop an abstract notion of naturalness based on information theory and use it to create a unifying framework for studying translations across the two axes. From their theory, it mathematically follows that the more one wants to increase the naturalness of a translation system, its accuracy must drop. This is a nice theoretical insight and is consistent with one's intuitions.

The authors also explore the utility of their theory empirically by examining outputs produced by a variety of systems. This leads to insights that were not possible to derive using prior frameworks.

**Reasons To Reject:**

I am most concerned with the heavy use of automatically generated translations in the empirical study (section 5.1). I would have preferred if the authors used
      1) publicly available MT and human outputs produced in WMT evaluations across several years: MT outputs created by older technology tended to produce more accurate but less fluent translations, while newer technology tends to produce more fluent translations. Showing that the framework introduced in this paper yields accuracy and naturalness score that are consistent with the above observation would strengthen the submission;
       2) a broader set of translation references, such as that introduced by Dreyer and Marcu (https://aclanthology.org/N12-1017/); comparing the accuracy and naturalness scores across millions of references translations would yield additional insights.

However, I find the results sufficient and convincing for the submission to be accepted at COLM'25. Follow-up work can address my concerns. And the community would benefit from learning and debating these findings now and developing public evaluation software/scripts of higher utility.

---

> ### Author Response · Authors · 2025-05-30
>
> Thank you for your highly positive review! Indeed, the primary aim of our paper is to spark debate and prompt researchers to reexamine how they evaluate their systems in the future. We are delighted that you are so receptive to this goal!
>
> Regarding your concerns about the empirical evaluation, we generally agree that there are several ways in which it could be improved, including the ideas you mention. Of course, having as tight an approximation to the true curve as possible would be of immense practical value. However, at present, this requires significant resources, and as such, we leave this for future work.

---

### Official Review · Reviewer_AhdH · 2025-05-12

**Rating:** 6
**Confidence:** 4
**Ethics Flag:** 1

**Summary:**

This submission proposes a mathematical framework for modeling the trade-off between accuracy and naturalness in Machine Translation. Some empirical results are shown to illustrate how it relates to actual MT evaluation results, using WMT24 data.

The paper is well structured, well written and overall quite clear, even in the more theoretical sections. The described framework is inspired by Blau & Michaeli's work on distortion-preception trade-off in image processing. The extension to MT is smart and well down, and well-connected to the various types of metrics used in the field.

It is unclear how this impacts MT research and evaluation, however. Although the distinction between naturalness and e.g. fluency is interesting and clearly described, such trade-offs have been well-known in the field for a while, as clearly stated in several places in the manuscript. This work shines a new light on this known problem that had received little theoretical (at least mathematical) work. Two outstanding issues remain:
1. Synthetic results (Sec. 5.1) yield outcomes that (reassuringly but unsurprisingly) match theory well. On the other hand, practical evaluation results (Fig. 2 and 3) do not appear to fit the theory particularly well. Authors argue that the trade-off curve only becomes relevant for better-performing models. Fair point, however in all examples, everything seems quite far from the curve, and it looks like essentially optimizing to the upper-right corner is the way to go. (This is indirectly acknowledged on lines 305-306.)
2. The practical impact for the MT community is unclear. If this trade-off changed the way data compression is evaluated and done, there does not seem to be a clear way to achieve that in MT. Evaluating along multiple dimensions has been tried before (again as rightly pointed out here), and the community has mostly move to one-dimensional metrics that are believed (or hand-waved) to optimize both accuracy and naturalness. It is unclear how this paper would change the state of things.

Note that I did not read the entirety of the massive appendix. There may be further discussions or results there that address my concerns.

**Questions To Authors:**

* l. 167, 240: I realize this may be a personal preference: style guides tend to advise against having a single subsection within a section.
* l. 178, "By definition ... perfectly accurate": do you mean "optimally" accurate, given Q*? There is no guarantee that the translation system is capable of perfect translation.
* l. 212, "convex in the first slot": assuming this means D(Q,P) is convex in Q?
* l. 216: : "convexity of A(N):" should arguably be concavity.
* l. 255: I assume this is convex in Q as per assumptions -- can you clarify it?
* l. 263: Why is the perfect score zero, is that an assumption or convenience choice with no loss of generality?
* l. 355-356: Isn't this suggestion of optimizing non-reference metrics (perplexity) essentially how neural models (NMT & LLM) work anyway?

**Reasons To Accept:**

This is an interesting new view on a previously known trade-off, with a neat mathematical framework, and clear relevance in the age of NMT and LLM-based translation.

**Reasons To Reject:**

The adequacy-fluency trade-off (and related) has been well-known in MT for a while, as acknowledged in various places in the paper. Beyond the neat mathematical framework, it is unclear what the practical impact of this work is, especially as the empirical results do not seem to agree well with theory. Some interesting lines of research are suggested but mainly left for future work.
Finally, while interesting to Machine Translation researchers, it is not clear that this work will have widespread interest for the rest of the COLM crowd, even for other NLP applications.

---

> ### Author Response · Authors · 2025-05-30
> **Rebuttal I: Addressing the main weaknesses you identify**
>
> Thank you for your nice review and the many engaging questions. We are delighted that you found our theoretical work interesting and insightful. We address your comments and concerns below.
>
> > "...practical evaluation results (Fig. 2 and 3) do not appear to fit the theory particularly well. Authors argue that the tradeoff curve only becomes relevant for better-performing models. Fair point, however in all examples, everything seems quite far from the curve…"
>
> This is a fair comment given the way we presented the results. However, please note that obtaining a precise estimate for the tradeoff curve is quite challenging and resource-intensive; therefore, we resorted to an approximation, as indicated in Section 5.1. It is difficult to know precisely how crude our approximation is; however, in Appendix F, we explain that the specific method we used is overly optimistic and makes the feasible region appear larger than it is. In this sense, the scatterplot corresponding to the evaluated WMT systems and the approximated A-N curve are not fully comparable.
>
> To get a tighter estimate for the curve, we would need to train a separate translation system for each point (or at least fine-tune a pre-trained model); this is how Blau and Michaeli obtained the curves in their work. While we believe this would be a valuable contribution, we leave it for future work.
>
> We believe the true curve is probably much closer to the plotted systems than the approximate curve, based on two observations: 1) the state-of-the-art translation systems are already really good, and 2) as we also mention as a concrete case study in the paper, in recent years, we consistently saw in WMT MT task evaluations that the winning submission according to automatic metrics (BLEU first, then neural metrics such as AutoRank from last year) are not the winners in human evaluations. We believe that our theory of the tradeoff explains this phenomenon.
>
> Indeed, we think the greatest value of Figures 2 and 3 is that they show the tradeoff between accuracy and naturalness of current systems and that it is worth concerning ourselves with evaluating along both axes.
>
> > "and it looks like essentially optimizing to the upper-right corner is the way to go."
>
> We don't agree with this conclusion. Indeed, the main point we hope to convey in the paper is that there is no unique "way to go.” The approximated A-N curves already show that there isn't a unique optimum that we can achieve by optimizing to the upper-right corner, but rather we need to consider a Pareto frontier of solutions.
>
> We can obtain different, Pareto-efficient (i.e., undominated) solutions by training a translation system using the average of the Lagrange dual objective in Eq. (5) with various settings for the slack variable $\beta$.
>
> > "The practical impact for the MT community is unclear. If this tradeoff changed the way data compression is evaluated and done, there does not seem to be a clear way to achieve that in MT. Evaluating along multiple dimensions has been tried before (again as rightly pointed out here), and the community has mostly move to one-dimensional metrics that are believed (or hand-waved) to optimize both accuracy and naturalness. It is unclear how this paper would change the state of things. "
>
> We believe that you are slightly conflating evaluation and optimization. We do **not** argue against optimizing neural (or any other) metrics. Instead, the point of our paper is to alert practitioners to the fact that there does not exist a single metric that can be optimized to attain optimal accuracy and perfect naturalness simultaneously in a practically meaningful way. This theoretical result directly contradicts a fairly widely held, yet formally unjustified, intuition in the community.
>
> As such, we hope that the paper will have a similar effect on the MT community that Blau and Michaeli's work did on the data compression community: researchers will stop looking for a "golden metric" to optimize and will have a more informed view of the inevitable accuracy-naturalness tradeoff of their systems.
>
> Furthermore, unlike in the data compression community, there has been much less research on statistical distances for evaluating translation systems. There have been some attempts, such as the Frechet embedding distance, the counterpart of the widely popular Frechet inception distance (FID) in image quality evaluation, but it is nowhere near as popular or even well-known. Thus, we hope our work will spark a new line of research into text naturalness evaluation techniques.
>
> Of course, we also hope that with our clarified view of accuracy and naturalness, which formalizes and illuminates all previous informal attempts, such as the adequacy-fluency tradeoff, the community will return to evaluating systems along two axes.

---

> ### Author Response · Authors · 2025-05-30
> **Rebuttal II: Relevance to COLM and Miscellaneous Issues**
>
> ## Relevance to COLM crowd
>
> Naturally, our work is motivated by and primarily targeted at the machine translation audience. Nonetheless, we believe our work fits well with the conference's topics. Concretely, point (3) in the list of topics of interest for COLM calls for submissions that are:
>  - *All about evaluation: benchmarks, simulation environments, scalable oversight, evaluation protocols and metrics, human and/or machine evaluation*
>
> In addition to the possible future directions we outlined earlier in our response, our work should be of interest to the broader COLM community and could serve as a basis for exploring interesting future directions. We foresee two particularly relevant future directions:
>  - For the case of text summarisation, the accuracy-naturalness tradeoff could be extended along a third axis: brevity. Then, this could be put in correspondence with the rate-distortion-perception tradeoff [1].
>  - Our new statistical distance could be used to detect unnatural / LLM-generated texts, an area that is already gaining significant popularity.
>
> ## Miscellaneous
> > l. 178, "By definition ... perfectly accurate": do you mean "optimally" accurate, given Q*? There is no guarantee that the translation system is capable of perfect translation.
>
> Yes, it should be optimally accurate. We will update this in our manuscript.
>
> > l. 212, "convex in the first slot": assuming this means D(Q,P) is convex in Q?
>
> Correct!
>
> > l. 216: : "convexity of A(N):" should arguably be concavity.
>
> Yes, nice catch! Thank you.
>
> > l. 255: I assume this is convex in Q as per assumptions -- can you clarify it?
>
> Indeed. We will provide a corresponding proof of this fact in Appendix E.
>
> > l. 263: Why is the perfect score zero, is that an assumption or convenience choice with no loss of generality?
>
> It's a convenience assumption. Indeed, you can see that this assumption is equivalent to assuming that every critic assigns a higher average score to any unnatural text. While this might sound intuitively true at first, we don't think that it should hold in practice. However, we also don't think that it should cause too significant a difference in practice. We made this assumption to simplify the presentation.
>
> > l. 355-356: Isn't this suggestion of optimizing non-reference metrics (perplexity) essentially how neural models (NMT & LLM) work anyway?
>
> Yes! Indeed, this is also why we used Gemma for the evaluation in Section 5!
>
> ## References
> - [1] Blau, Y., & Michaeli, T. (2019, May). Rethinking lossy compression: The rate-distortion-perception tradeoff. In International Conference on Machine Learning (pp. 675-685). PMLR.

---

> > ### Comment · Reviewer_AhdH · 2025-06-09
> >
> > First I would like to thank the authors for this stimulating work, and for their substantial and well-thought out replies to reviewers. For what it is worth, I fully respect that it is entirely your prerogative to present this work at the venue of your choice. I should not have placed this comment re. "relevance to COLM" in the "Reasons to reject" section.
> >
> > After reading the substantial comments from other reviewers and extensive author replies, I now lean towards acceptance. There is still one key factor reducing my enthusiasm re. this submission. In my understanding:
> > * The author introduce and define the concept of naturalness and convincingly illustrate how it differs from fluency (e.g. end Sec. 3.1);
> > * They demonstrate a tradeoff between accuracy and naturalness, and crucially show that it is not possible to jointly optimize both.
> >
> > On the other hand, the MT community has long focused on the adequacy-fluency trade-off. The relevance and impact of this work hinges on the fact that either the community switches to accuracy-naturalness, or that a similar trade-off exists for adequacy-fluency. I understand that the authors clearly distance from the latter, and the former seems far from given. Prior studies on translationese, how to detect it and how it impacts MT, would arguably suggest that the choice between fluency and naturalness has been considered and internalized by the community, acknowledging the limits of naturalness.

---

> > > ### Author Response · Authors · 2025-06-10
> > >
> > > Thank you very much for raising your score!
> > >
> > > With respect to fluency-adequacy, this is a fair point and one that we discussed during the development of our work. In fact, we started working using those terms, but in the discussion with colleagues we soon figured out that due to its long tradition, these concepts had strong connotations, which in turn were different for different people, and a formalization that would fit them all would be difficult, if not impossible. We found that the definition adapted from information theory was a good fit for "naturalness", as we are basically assessing if the outputs originate from a "natural" distribution, and we decided to move forward with it. In our opinion, the two concepts are closely related. Can a sentence that is not fluent be considered natural? With the advent of social networks and other forms of quick communication we might need to revisit this question, but the advantage of our theory is that by choosing the appropriate reference distribution we can account for these variations. And in
> > > fact, if we choose a distribution of "strictly fluent" sentences we should be covering the "traditional" case.
> > >
> > > Adequacy is probably less controversial, and we think that it clearly can be considered an instance of an accuracy metric. But again, we tried to avoid giving a (possibly controversial) definition of adequacy in the traditional sense and explicitly kept to the more general concept of accuracy.

---

### Official Review · Reviewer_Z5DR · 2025-05-13

**Rating:** 7
**Confidence:** 3
**Ethics Flag:** 1

**Summary:**

This paper explores the question of tradeoffs between accuracy and naturalness in machine translation. The paper formalizes a definition of naturalness based on target language text corpora and demonstrates that under that definition and their definition of accuracy, there will generally be a tradeoff between accuracy and naturalness, resulting in a pareto frontier that translation systems may approach. They show some well-known results (correlation between adequacy and fluency at lower quality levels) and also provide evidence related to recent questions about whether there may be less correlation between the two as translation quality improves. This work provides a formalization that may be useful in considering how machine translation systems are trained and evaluated, though in practice there are likely to be many questions and debates about how best to implement it (e.g., what constitutes an appropriate corpus for measuring naturalness, what metrics are appropriate for measuring adequacy).

**Questions To Authors:**

- The choice of the term “reference metric” has the potential to be confusing, in part because it is defined to include QE metrics, which are sometimes referred to as reference-free or referenceless metrics.
- Can you comment on the interplay between naturalness and topics or domains? There is some discussion in the paper of the idea that there may be certain domains where it may be preferred to lean towards accuracy (e.g., scientific papers) vs. naturalness (e.g., literature). What about situations where you are considering translations of things that are very specific to the source language (e.g., specific cultural or political topics, or even regional or geographic topics) that are very different from or very infrequently discussed in the target language? In such a situation, in what ways do you expect (or not expect) these to be captured in terms of naturalness in the ““true” monolingual reference over the target language”?
- Lines 163-166 state that the “proposed theory has no notion of “sentence-level” naturalness”. This is the first time that sentence-level questions are brought up, and it is not clear where this is coming from, how it should be interpreted, or how it connects to the next sentences. This could be expanded or rephrased.
- It would be appropriate to cite prior work on training and/or decoding machine translation systems with multiple objectives, e.g., https://aclanthology.org/P12-1001.pdf (While this focuses on earlier MT paradigms, it touches on many of the same issues raised here.) and more recently https://proceedings.neurips.cc/paper_files/paper/2021/file/79ec2a4246feb2126ecf43c4a4418002-Paper.pdf

**Reasons To Accept:**

- The definition of naturalness based on a corpus rather than through an attempt to define a prescriptive definition of naturalness provides quite a bit of flexibility. This neatly sidesteps some thorny issues, though they may reappear when one attempts to implement this in practice.
- The paper makes good and appropriate use of existing data for the evaluation, which makes it reproducible.
- The paper provides a thorough definition, description, and formalization of its approaches.

**Reasons To Reject:**

- Line 129 mentions that condition 2 (that the function $\Delta$ is minimized when $y^c$ equals $y^r$) is difficult to prove for neural metrics. This seems like a potentially risky understatement, given that some neural metrics are known to have pathological errors such as “universal translations” that score high regardless of the reference (https://aclanthology.org/2023.acl-long.297.pdf).
- More generally, there is a bit of handwaving about what exactly various neural metrics are measuring, which in some ways brings us back to the original underlying question of how possible it is to fully treat adequacy and fluency as two completely separate dimensions of quality.

---

> ### Author Response · Authors · 2025-05-30
>
> Thank you for your nice review and your constructive critique! We address your concerns below.
>
> ## Worry about neural metrics being true distortion metrics
>
> This is a fair concern, and we certainly worried about things along the same lines. As you mention, neural metrics are not "adversarially robust" in that they can be tricked with universal translations. However, neural metrics certainly perform well in the "average case." Thankfully, the average case is what ultimately matters for our theory since we define accuracy in terms of the *expected* distortion. Indeed, if neural metrics weren't good in the average case, then they wouldn't have much use.
>
> Nonetheless, getting better neural metrics that are more adversarially robust is an important future direction for MT research and such robust metrics in turn be more reliable to be used in our evaluation framework too.
>
> ## Handwaving about what neural metrics measure
>
> While this is, in fact, not an issue for the theory we develop, it is an excellent point that also arose during our discussions with colleagues while writing the paper. Indeed, in the community there is an unwritten feeling that neural metrics place a heavier weight on how well a particular translated sentence sounds than on the accuracy of the translation. As a side note, in image compression, there also exist "perceptual distortion metrics" that aim to capture differences that matter to the human visual system.
>
> However, importantly, so long as there is an element of comparison with a reference, and assuming that our assumptions are satisfied, neural metrics are still reference metrics, and the theory applies.
>
> In other words, we are **not** advocating against optimizing systems using neural metrics. We are merely pointing out that if we optimize a system for a neural metric versus a "pure accuracy metric" such as BLEU or chrF, we will end up on a different part of the accuracy-naturalness curve since BLEU prioritizes accuracy over a neural metric that also measures naturalness.
>
> ## Choice of the term reference metric
>
> We agree that it is confusing, and we have given careful consideration to our choice of terminology. We ended up choosing this terminology for two reasons: 1) our use of the term aligns with its use in the information theory / learned data compression literature, which is the original inspiration of the work; 2) as we point out in the paper, QE metrics do use the source sentence as a reference. One could argue that QE metrics should be called "target reference-free" metrics.
>
> > "Can you comment on the interplay between naturalness and topics or domains?…"
>
> This is a great question, and it really pushes the boundaries of the theory. We briefly discuss this on lines 219-228, but let us be more explicit here. We specifically believe that the appropriate behaviour lies on a spectrum, with the following two approaches as its extremes:
>  - We produce a full explanation
>  - We either mirror-translate the term or leave it as is.
>
> The appropriate approach depends on the context. For example, when interacting with a user who is more familiar with internet slang, we might wish to translate the English word "doomscrolling" as "doomscrolling." However, for someone less familiar, we might want to produce a translation that explains the meaning of doomscrolling.
>
> Note that currently, there is no method available to assess this, and it is a critical direction for future work.
>
> ## Regarding sentence-level naturalness
>
> That is a fair point; we just meant to emphasize that in our theory, naturalness is a property of distributions and not of individual segments. In other words, what makes sense to talk about is whether a "system" sounds natural, but it doesn't make sense to say that individual segments are natural or not from our theory's perspective. Of course, one can still debate sentence-level naturalness outside of our theory. We will clarify this in the camera-ready version of our paper.
>
> ## Cite prior work
> Thank you for the references! We will cite them in the updated version of the manuscript!

---

> > ### Comment · Reviewer_Z5DR · 2025-06-05
> >
> > Thank you for your reply. I think the paper will benefit from incorporating some of these clarifications. In particular, I think both the sentence-level naturalness explanation and the terminology question around reference metrics would benefit from at least a solid footnote, if not main-body text. Just being very clear about why you're using the terminology (and acknowledging how there's a bit of a terminology collision with other work) will help your readers adjust to this usage.
> >
> > In a case like "doomscrolling" there's a very natural and concise way to express this to someone who is familiar with this term. For someone wholly unfamiliar with the concept, it may require lengthy explicitation. This may be too tangential to your main points, but it may help readers to better understand your concepts of naturalness: would a lengthy but fluent explicitation of an unknown concept be considered "natural"? In some evaluations of translations (and using particular definitions of naturalness), such explicitation may be considered awkward or less natural.

---

### Official Review · Reviewer_LQ9z · 2025-05-13

**Rating:** 8
**Confidence:** 4
**Ethics Flag:** 1

**Summary:**

The paper extends the work on distortion-perception trade-off to the field of (machine) translation. The paper tries to provide a rigorous mathematical justification for the trade-off between translation accuracy and naturalness, proposing that these criteria are fundamentally at odds. The work contains a number of diverse contributions:

* A theoretical extension of the distortion-perception trade-off to translation.
* Empirical validation of the hypothesis.
* Extensive supplemental material containing rigorous mathematical derivations and examples.

The work is ambitious and in my opinion exceeds the scope of a conference paper.

**Questions To Authors:**

* Why did you select a relatively narrow domain (NewsCrawl) for estimation of $R_y$? I realize this is the domain of the translated data, however, it doesn't necessarily represent the full spectrum of "natural-sounding German".

**Reasons To Accept:**

* Scope of the work; the authors combine mathematical work and extensive empirical evaluation at a scale that is rarely seen in conference papers.
* Thought-provoking results. The distinction between adequacy/accuracy ("Translation model") and naturalness/fluency ("Language model") is not new, however the statement that as translation quality increases, they become fundamentally at odds with one another, seems like a very interesting result which can spark debate in the community.
* I would highlight the synthetic result which shows convincingly the trade-off at the Pareto frontier (illustrated in Figure 1).
* Clear writing; the paper is easy to follow and explains all the concepts well.

**Reasons To Reject:**

I have doubts about how well the empirical results (specifically the analysis of WMT system submissions) actually fit the hypothesis.
* In Figure 2, there is no clear sign of a trade-off. In fact, better-performing systems appears to be closing in on the Pareto frontier in a fairly straight line, improving both accuracy and naturalness.
* In Figure 3, each language pair paints a completely different picture. I don't observe any systematic "correlation far away from" and "anti-correlation close to" the Pareto frontier and I doubt it can be reliably inferred statistically, given the small sample sizes.

Is it possible that while the trade-off exists _in theory_, in practice we are actually not close to the "quality limit" and we can still continue improving both aspects?

As a separate issue, I have some doubts about the assumptions formulated in sections 2 and 3.

The definitions state that sentence pairs are drawn from the joint distribution $P(x,y)$. If that's so, I don't understand how it's possible to assume that this leads to a "natural" $P(x)$ (after marginalization, i.e. $P(x) = \sum_{y} P(x,y)$) but a "translation-biased" $P(y)$  (after marginalization, i.e. $P(y) = \sum_{x} P(x,y)$ ). Concretely, the authors state that:

> For the purposes of this paper, we think of Px as a “true” monolingual distribution over the source language and Py|x as the distribution of human translations into the target language given the source sentence x. Therefore, since we obtain Py by marginalising over the input, it is the distribution over “human translations into the target language.”

I believe we can either assume some sort of sampling bias which skews $P_{x,y}$, but then both $P_x$ and $P_y$ are affected. Alternatively, we assume that we can get a correct estimate of $P(x,y)$ but then the estimates of _both_ marginal probability distributions should be correct as well? (As in, English-German parallel sentences are neither natural English, nor natural German. Or they are both natural. But I don't see why one side is natural and the other one isn't. If the assumption is indeed that all the sentences come from "original language" and their target-language version are always translations, isn't that flawed? Parallel corpora will typically contain a mix of both, and each has very different characteristics in practice.)

There is also the underlying related assumption that "starting from a different source language leads to a different $P(y)$". How is this mathematically justified? I can relate to the examples and would tend to agree that this occurs _in practice_, I just don't follow the chain of thought here.

Maybe I'm misunderstanding Appendix B but if we assume either non-invertibility (like the original paper) or non-determinism, would we still need these assumptions?

Again, I don't necessarily doubt that this occurs in practice, I just think there's a flaw in the "pure mathematical argument" in the main text.

Finally, going back to the empirical results, there are some relatively strong assumptions about MQM annotations which I would prefer to see validated first: (a) fluency score doesn't depend on the source, even though annotators have access to the source sentence (b) expected fluency of a natural sentence in the target language is perfect (0 errors).

---

> ### Author Response · Authors · 2025-05-30
>
> Thank you for your nice review and especially for your detailed critique. We address your concerns point-by-point below.
>
> ## Doubts about empirical results fitting the hypothesis. Could we still be far from the curve?
> This is a fair concern, and we struggled with the tradeoff (pun intended) between clearly illustrating our theory and the tightness of the presented empirical results.
>
> The main issue is that producing a tighter estimate of the curve is extremely challenging computationally: we would need to optimise a language model using the expectation of the Lagrange dual from Eq (5) for each accuracy-naturalness tradeoff point. Indeed, this method is how Blau and Michaeli obtained their corresponding tradeoff curves for image restoration in their original paper.
>
> Instead, we opted for the more feasible alternative, which is to maximise the one-shot Lagrange dual in Eq (5) over a large number of candidate translations for each source sentence. However, as we note at the end of Section 5.1 and explain in detail in Appendix F, this approach presents an overly optimistic estimate of the curve, making the feasible region appear larger than it actually is.
>
> It is difficult to know how far we are from the curve; we could be far. However, we believe the true curve is probably much closer to the plotted systems than the approximate curve, based on two observations: 1) the state-of-the-art translation systems are already really good and 2) as we also mention as a concrete case study in the paper, in recent years, we consistently saw in WMT MT task evaluations that the winning submission according to automatic metrics (BLEU first, then neural metrics such as AutoRank from last year) are not the winners in human evaluations. We believe that our theory of the tradeoff explains this phenomenon.
>
> > "In Figure 3, …  I don't observe any systematic "correlation far away from" and "anti-correlation close to” ... I doubt it can be reliably inferred statistically..."
>
> We also agree that it is probably very hard to estimate the "transition point" where the two quantities start to anti-correlate. Our argument only demonstrates that a transition point must always exist. However, we believe the existence result is already a valuable contribution to the literature.
>
> ## Doubts about the theoretic assumptions
> Our theory is based on and reflects current practices in machine translation: the source language is typically derived from a monolingual corpus, which professional translators then translate into the target language. Indeed, there has been a recent explicit effort in the MT community to avoid back-translations or translations in both directions [1, 2].
>
> Philosophically, the issue we had is that defining a joint distribution of perfectly natural source-target language pairs seems unsound/troublesome. Practically, the issue is that parallel distributions are "direction-dependent:" Translating an English corpus into German will inevitably yield a different "joint distribution" compared to translating German into English.
>
> However, our theory sidesteps this issue, and the setup only has one crucial feature: we do not necessarily assume that the target marginal $P_y$ represents the distribution of "perfectly natural texts." Instead, it is up to the practitioner to identify an appropriate monolingual reference distribution $R_y$. This assumption is a key differentiating factor between Blau & Michaeli's work and ours.
>
> ## Worries about MQM score conversion and interpretation
> We are pleased that you have identified these concerns; we had considered them as well but did not have the space in the submission to discuss them. We address your two comments below.
>
> > "fluency score doesn't depend on the source, even though annotators have access to the source sentence"
>
> Of course, as current MQM evaluations are conducted, the fluency scores could depend on the source text. While this is undoubtedly an issue that needs to be addressed in future evaluations, we do not think that it should have too significant an effect.
>
> > "expected fluency of a natural sentence in the target language is perfect"
>
> This is the more concerning of the two issues. Indeed, this assumption is equivalent to assuming that every critic assigns a higher average score to any unnatural text. While this might sound intuitively true at first, we don't think that it should hold in practice. However, we also don't think that it should cause too significant a difference in practice.
>
> ## Why did we use monolingual NewsCrawl?
> We used it simply to illustrate our point, and we wish to make no claims as to whether it represents the full spectrum of natural-sounding German.
>
> ## References
> 1. Toral, A. et al. (2018). Attaining the unattainable? Reassessing claims of human parity in neural machine translation. arXiv.
> 2. Läubli, S et al. (2020). A set of recommendations for assessing human–machine parity in language translation. JAIR

---

> > ### Comment · Reviewer_LQ9z · 2025-06-10
> >
> > Thank you for the response. I've decided to keep the current (positive) overall rating.

---

### Author Response · Authors · 2025-05-30

We thank the reviewers for their thoughtful and constructive reviews. We are delighted that they found our paper thought-provoking. Indeed, sparking a debate within machine translation and related communities about model evaluation practices is one of our top aims. Furthermore, we were also happy that they found our theoretical results interesting and our writing clear.

The single greatest worry of the reviewers was the alignment of the empirical results with the theory, especially in Figures 2 and 3. We agree that it is not immediately clear from the figures alone that current state-of-the-art systems should already be suffering from the accuracy-naturalness tradeoff.

However, we have two responses:
1. Even if current systems are not yet suffering from the tradeoff, Figure 2 demonstrates a clear (overly optimistic) boundary for what accuracy-naturalness tradeoffs are achievable and how much more we can expect future systems to improve compared to the state-of-the-art. Conversely, the figure clearly shows that there is a non-trivial region of unachievable (accuracy, naturalness) pairs. In particular, the “top-right corner is unachievable,” meaning that there is no system that simultaneously achieves perfect accuracy and perfect naturalness. Hence, a fortiori, there is also no method to obtain such a system, e.g. by optimising a system using some as yet undiscovered “golden objective”.
2. As we explain in Appendix F, our approximation is overly optimistic, meaning that the curves shown in Figure 2 make the feasible region appear larger than it actually is. Indeed, we believe that, in reality, the true curves are much closer to the systems plotted in Figures 2 and 3. Our intuition is based on two observations: 1) current translation systems are already quite good, and 2) as we note in the paper, there have been several recent, well-documented cases where humans did not prefer a system that performed best according to some automated metric.

---

### Decision · Program_Chairs · 2025-07-08

**Decision:**

Accept

**Comment:**

The paper posits that in MT there is an accuracy-naturalness tradeoff and (calling back to existing work on the perception-distortion tradeoff, Blau & Michaeli) shows that in theory both cannot be perfectly satisfied at the same time, plus showing in practice how current models do on either objective, not contradicting the theory (though not a stellar endorsement).

Reviewer LQ9z praises the super-conference-level scope of the work and its ability to spark discussion on known concepts, appreciating writing and theory, but also noting a disconnect to the practical results as WMT systems seem very far from the theoretical limits so the proposed behavior near that limit cannot be empirically observed (authors rebut that the theoretical limits are overly optimistic due to computational constraints in their calculation). (They also take issue with the formulation of bitext joint distributions, but I would side with the authors here in that indeed the marginals of such an empirical joint distribution are not created equally.)

Reviewer Z5DR in addition praises the corpus-based naturalness definition, use of existing data, and thorough definitions. They note pathological cases in neural metrics like the ones used here, though the authors believably rebut that behavior in expectation is good enough for their theory.

Reviewer AhdH in addition wonders how this paper could have impact in MT given the community's preference for single metrics over an evaluation along multiple axes---and whether it fits well at COLM (which authors rebut, but I see the reviewer's point, but I really like the paper, so I would argue for it fitting in COLM anyway). They also bring up the question of how accuracy-naturalness can relate to the adequacy-fluency tradeoff that the MT community has chosen to internalize more, but the authors say this simply came down to not wanting to deal with heavy connotation baggage.

Reviewer z3nj also very much appreciates the paper, but adds criticism about the use of automatic translation in the evaluation over existing human and machine translations, especially given that older systems may illuminate different areas of the plane.

In summary, the paper is clearly of high quality and clarity, with sufficient originality and relevance to the community. I personally really like the way it is set up and agree that it should prove to be a very stimulating and interesting paper, even if mildly disconnected from the current reality of systems.